# Conservative Dual Policy Optimization for Efficient Model-Based Reinforcement Learning

**Shenao Zhang**
Georgia Institute of Technology
Atlanta, GA 30332
shenao@gatech.edu

## Abstract

Provably efficient Model-Based Reinforcement Learning (MBRL) based on optimism or posterior sampling (PSRL) is ensured to attain the global optimality asymptotically by introducing the complexity measure of the model. However, the complexity might grow exponentially for the simplest nonlinear models, where global convergence is impossible within finite iterations. When the model suffers a large generalization error, which is quantitatively measured by the model complexity, the uncertainty can be large. The sampled model that current policy is greedily optimized upon will thus be unsettled, resulting in aggressive policy updates and over-exploration. In this work, we propose *Conservative Dual Policy Optimization* (CDPO) that involves a *Referential Update* and a *Conservative Update*. The policy is first optimized under a reference model, which imitates the mechanism of PSRL while offering more stability. A conservative range of randomness is guaranteed by maximizing the expectation of model value. Without harmful sampling procedures, CDPO can still achieve the same regret as PSRL. More importantly, CDPO enjoys monotonic policy improvement and global optimality simultaneously. Empirical results also validate the exploration efficiency of CDPO.

## 1 Introduction

Model-Based Reinforcement Learning (MBRL) involves acquiring a model by interacting with the environment and learning to make the optimal decision using the model [55, 32]. MBRL is appealing due to its significantly reduced sample complexity compared to its model-free counterparts. However, greedy model exploitation that assumes the model is sufficiently accurate lacks guarantees for global optimality. The policies can be suboptimal and get stuck at local maxima even in simple tasks [10].

As such, several provably-efficient MBRL algorithms have been proposed. Based on the principle of *optimism in the face of uncertainty* (OFU) [56, 49, 10], OFU-RL achieves the global optimality by ensuring that the optimistically biased value is close to the real value in the long run. Based on Thompson Sampling [62], Posterior Sampling RL (PSRL) [57, 42, 43] explores by greedily optimizing the policy in an MDP which is sampled from the posterior distribution over MDPs. Beyond finite MDPs, to obtain a general bound that permits sample efficiency in various cases, we need to introduce additional complexity measure. For example, [49, 43] provide an $\widetilde{O}(\sqrt{d_E T})$ regret for both OFU and PSRL with eluder dimension $d_E$ capturing how effectively the model generalizes. However, it is recently shown [13, 33] that the eluder dimension for even the simplest nonlinear models cannot be polynomially bounded. The effectiveness of the algorithms will thus be crippled.

The underlying reasons for such ineffectiveness are the aggressive policy updates and the over-exploration issue. Specifically, when a nonlinear model is used to fit complex transition functions, its generalizability will be poor compared to simple linear problems. If a random model is selected from the large hypothesis, e.g., optimistically chosen or sampled from the posterior, it is "unsettled".

36th Conference on Neural Information Processing Systems (NeurIPS 2022).

In other words, the selected model can change dramatically between successive iterations. Policy updates under this model will also be aggressive and thus cause value degradation. What's worse, large epistemic uncertainty results in an unrealistic model, which drives agents for uninformative exploration. An exploration step can only eliminate an exponentially small portion of the hypothesis.

In this work, we present *Conservative Dual Policy Optimization* (CDPO), a simple yet provable MBRL algorithm. As the sampling process in PSRL harms policy updates due to the unsettled model during training, we propose the *Referential Update* that greedily optimizes an intermediate policy under a *reference model*. It mimics the sampling-then-optimization procedure in PSRL but offers more stability since we are free to set a steady reference model. We show that even without a sampling procedure, CDPO can match the expected regret of PSRL up to constant factors for any proper reference model, e.g., the least squares estimate where the confidence set is centered at. The *Conservative Update* step then follows to encourage exploration within a reasonable range. Specifically, the objective of a reactive policy is to maximize the *expectation* of model value, instead of a single model's value. These two steps are performed in an iterative manner in CDPO.

Theoretically, we show the statistical equivalence between CDPO and PSRL with the same order of expected regret. Additionally, we give the iterative policy improvement bound of CDPO, which guarantees monotonic improvement under mild conditions. We also establish the sublinear regret of CDPO, which permits its global optimality equipped with any model function class that has a bounded complexity measure. To our knowledge, the proposed framework is the first that *simultaneously* enjoys global optimality and iterative policy improvement. Experimental results verify the existence of the over-exploration issue and demonstrate the practical benefit of CDPO.

## 2 Background

### 2.1 Model-Based Reinforcement Learning

We consider the problem of learning to optimize an infinite-horizon $\gamma$-discounted Markov Decision Process (MDP) over repeated episodes of interaction. Denote the state space and action space as $\mathcal{S}$ and $\mathcal{A}$, respectively. When taking action $a \in \mathcal{A}$ at state $s \in \mathcal{S}$, the agent receives reward $r(s,a)$ and the environment transits into a new state according to probability $s' \sim f^*(\cdot|s,a)$. Here, $f^*$ is a dirac measure for deterministic dynamics and is a probability distribution for probabilistic dynamics.

In model-based RL, the true dynamical model $f^*$ is unknown and needs to be learned using the collected data through episodic (or iterative) interaction. The history data up to iteration $t$ then forms $\mathcal{H}_t = \{\{s_{h,i}, a_{h,i}, s_{h+1,i}\}_{h=0}^{H-1}\}_{i=1}^{t-1}$, where $H$ is the actual timesteps agents run in an episode. The posterior distribution of the dynamics model is estimated as $\phi(\cdot|\mathcal{H}_t)$. Alternatively, the frequentist model of the mean and uncertainty can also be estimated. Specifically, consider the model function class $\mathcal{F} = \{f : \mathcal{S} \times \mathcal{A} \to \mathcal{S}\}$ with size $|\mathcal{F}|$, which contains the real model $f^* \in \mathcal{F}$. The confidence set (or model hypothesis set) $\mathcal{F}_t \subset \mathcal{F}$ is introduced to represent the range of dynamics that is statistically plausible [49, 43, 10]. To ensure that $f^* \in \mathcal{F}_t$ with high probability, one way is to construct the confidence set as $\mathcal{F}_t := \{f \in \mathcal{F} \mid \|f - \widehat{f}_t^{LS}\|_{2,E_t} \leq \sqrt{\beta_t}\}$. Here, $\beta_t$ is an appropriately chosen confidence parameter (via concentration inequality), the cumulative empirical 2-norm is defined by $\|g\|_{2,E_t}^2 := \sum_{i=1}^{t-1} \|g(x_i)\|_2^2$. The least squares estimate is

$$\widehat{f}_t^{LS} := \operatorname*{argmin}_{f \in \mathcal{F}} \sum_{(s,a,s') \in \mathcal{H}_t} \|f(s,a) - s'\|_2^2. \tag{2.1}$$

Denote the state and state-action value function associated with $\pi$ on model $f$ by $V_\pi^f : \mathcal{S} \to \mathbb{R}$ and $Q_\pi^f : \mathcal{S} \times \mathcal{A} \to \mathbb{R}$, respectively, which are defined as

$$V_\pi^f(s) = \mathbb{E}\left[\sum_{h=0}^{\infty} \gamma^h r(s_h, a_h) \,\middle|\, s_0 = s, \pi, f\right], \quad Q_\pi^f(s,a) = \mathbb{E}\left[\sum_{h=0}^{\infty} \gamma^h r(s_h, a_h) \,\middle|\, s_0 = s, a_0 = a, \pi, f\right].$$

The objective of RL is to learn a policy $\pi^* = \operatorname{argmax}_\pi J(\pi)$ that maximizes the expected return $J(\pi)$. Denote the initial state distribution as $\zeta$. Under policy $\pi$, the state visitation measure $\nu_\pi(s)$ over $\mathcal{S}$ and the state-action visitation measure $\rho_\pi(s,a)$ over $\mathcal{S} \times \mathcal{A}$ in the true MDP are defined as

$$\nu_\pi(s) = (1-\gamma) \cdot \sum_{h=0}^{\infty} \gamma^h \cdot \mathbb{P}(s_h = s), \quad \rho_\pi(s,a) = (1-\gamma) \cdot \sum_{h=0}^{\infty} \gamma^h \cdot \mathbb{P}(s_h = s, a_h = a), \tag{2.2}$$

where $s_0 \sim \zeta$, $a_h \sim \pi(\cdot|s_h)$ and $s_{h+1} \sim f^*(\cdot|s_h, a_h)$. The objective $J(\pi)$ is then

$$J(\pi) = \mathbb{E}_{s_0 \sim \zeta}[V_\pi^{f^*}(s_0)] = \mathbb{E}_{(s,a) \sim \rho_\pi}[r(s,a)] \tag{2.3}$$

## 2.2 Cumulative Regret and Asymptotic Optimality

A common criterion to evaluate RL algorithms is the cumulative regret, defined as the cumulative performance discrepancy between policy $\pi_t$ at each iteration $t$ and the optimal policy $\pi^*$ over the run of the algorithm. The (cumulative) regret up to iteration $T$ is defined as:

$$\text{Regret}(T, \pi, f^*) := \sum_{t=1}^{T} \int_{s \in \mathcal{S}} \zeta(s)(V_{\pi^*}^{f^*}(s) - V_{\pi_t}^{f^*}(s)), \tag{2.4}$$

In the Bayesian view, the model $f^*$, the learning policy $\pi$, and the regret are random variables that must be learned from the gathered data. The Bayesian expected regret is defined as:

$$\text{BayesRegret}(T, \pi, \phi) := \mathbb{E}\left[\text{Regret}(T, \pi, f^*) \mid f^* \sim \phi\right]. \tag{2.5}$$

One way to prove the asymptotic optimality is to show that the (expected) regret is sublinear in $T$, so that $\pi_t$ converges to $\pi^*$ within sufficient iterations. To obtain the regret bound, the *width of confidence set* $\omega_t(s,a)$ is introduced to represent the maximum deviation between any two members in $\mathcal{F}_t$:

$$\omega_t(s,a) = \sup_{\underline{f}, \overline{f} \sim \mathcal{F}_t} \|\overline{f}(s,a) - \underline{f}(s,a)\|_2. \tag{2.6}$$

# 3 Provable Model-Based Reinforcement Learning

In this section, we analyze the central ideas and limitations of greedy algorithms as well as two popular theoretically justified frameworks: optimistic algorithms and posterior sampling algorithms.

**Greedy Model Exploitation.** Before introducing provable algorithms, we first analyze greedy model-based algorithms. In this framework, the agent takes actions assuming that the fitted model sufficiently accurately resembles the real MDP. Algorithms that lie in this category can be roughly divided into two groups: model-based planning and model-augmented policy optimization. For instance, Dyna agents [61, 20, 17] optimize policies using model-free learners with model-generated data. The model can also be exploited in first-order gradient estimators [18, 12, 9] or value expansion [15, 6]. On the other hand, model-based planning, or model-predictive control (MPC) [40, 41], directly generates optimal action sequences under the model in a receding horizon fashion.

However, greedily exploiting the model without *deep exploration* [45] will lead to suboptimal performance. The resulting policy can suffer from premature convergence, leaving the potentially high-reward region unexplored. Since the transition data is generated by the agent taking actions in the real MDP, the dual effect [4, 27] that current action influences *both* the next state and the model uncertainty is not considered by greedy model-based algorithms.

**Optimism in the Face of Uncertainty.** A common provable exploration mechanism is to adopt the principle of *optimism in the face of uncertainty* (OFU) [56, 49, 10]. With OFU, the agent assigns to its policy an optimistically biased estimate of virtual value by *jointly* optimizing over the policies and models inside the confidence set $\mathcal{F}_t$. At iteration $t$, the OFU-RL policy $\pi_t$ is defined as:

$$\pi_t = \operatorname*{argmax}_\pi \max_{f_t \in \mathcal{F}_t} V_\pi^{f_t}. \tag{3.1}$$

Most asymptotic analyses of optimistic RL algorithms can be abstracted as showing two properties: the virtual value $V_\pi^f$ is sufficiently high, and it is close to the real value $V_\pi^{f^*}$ in the long run. However, in complex environments where the generalizability of nonlinear models is limited, large epistemic uncertainty will result in an unrealistically large optimistic return that drives agents for uninformative exploration. What's worse, such suboptimal exploration steps eliminate only a small portion of the model hypothesis [13], leading to a slow converging process and suboptimal practical performance.

**Posterior Sampling Reinforcement Learning.** An alternative exploration mechanism is based on *Thompson Sampling* (TS) [62, 52], which involves selecting the maximizing action from a statistically

plausibly set of action values. These values can be associated with the MDP sampled from its posterior distribution, thus giving its name *posterior sampling for reinforcement learning* (PSRL) [57, 42, 43]. The algorithm begins with a prior distribution of $f^*$. At each iteration $t$, a model $f_t$ is sampled from the posterior $\phi(\cdot|\mathcal{H}_t)$, and $\pi_t$ is updated to be optimal under $f_t$:

$$f_t \sim \phi(\cdot|\mathcal{H}_t), \ \pi_t = \underset{\pi}{\operatorname{argmax}} \ V_\pi^{f_t}. \tag{3.2}$$

The insight is to keep away from actions that are unlikely to be optimal in the real MDP. Exploration is guaranteed by the randomness in the sampling procedure. Unfortunately, executing actions that are optimally associated with a single sampled model can cause similar over-exploration issues [52, 51]. Specifically, an imperfect model sampled from the large hypothesis can cause aggressive policy updates and value degradation between successive iterations. The suboptimality degree of the resulting policies depends on the epistemic model uncertainty. Besides, executing $\pi_t$ is not intended to offer performance improvement for follow-up policy learning, but only to narrow down the model uncertainty. However, this elimination procedure will be slow when the model suffers a large generalization error, which is quantitatively formulated in the model complexity measure below.

**Complexity Measure and Generalization Bounds.** In RL, we seek to have the sample complexity for finding a near-optimal policy or estimating an accurate value function. When given access to a generative model (i.e., an abstract sampling model) in finite MDPs, it is known that the (minimax) number of transitions the agent needs to observe can be sublinear in the model size, i.e. smaller than $O(|\mathcal{S}|^2|\mathcal{A}|)$. Beyond finite MDPs where the number of states is large (or countably or uncountably infinite), we are interested in the learnability or generalization of RL. Unfortunately, it is impossible for agnostic reinforcement learning that finds the best hypothesis in some given policy, value, or model hypothesis class: the number of needed samples depends exponentially on the problem horizon [24]. Despite of the structural assumptions, e.g. linear MDPs [66, 22, 65] or low-rank MDPs [21, 38], we focus on the generalization bounds that can cover various cases. This can be done with additional complexity measure, e.g. eluder dimension [49], witness rank [60], or bilinear rank [14].

By introducing the eluder dimension $d_E$ [49], previous work [43, 44] established regret $\widetilde{O}(\sqrt{d_E T})$ for both OFU-RL and PSRL. Intuitively, the eluder dimension captures how effectively the model learned from observed data can extrapolate to future data, and permits sample efficiency in various (linear) cases. Nevertheless, it is shown in [13, 33] that even the simplest nonlinear models do not have a polynomially-bounded eluder dimension. The following result is from Thm. 5.2 in Dong et al. [13] and similar results are also established in [33].

**Theorem 3.1** (Eluder Dimension of Nonlinear Models [13])**.** The eluder dimension $dim_E(\mathcal{F}, \varepsilon)$ (c.f. Definition 5.6) of one-layer ReLU neural networks is at least $\Omega(\varepsilon^{-(d-1)})$, where $d$ is the state-action dimension, i.e. $(s,a) \in \mathbb{R}^d$. With more layers, the requirement of ReLU activation can be relaxed.

As a result, additional complexity is hidden in the eluder dimension, e.g. when we choose $\varepsilon = T^{-1}$, regret $\widetilde{O}(\sqrt{d_E T})$ contains $d_E = \Omega(T^{d-1})$ and is no longer sublinear in $T$. In this case, previous provable exploration mechanisms will lose the desired property of global optimality and sample efficiency, which is the underlying reason for the over-exploration issue.

# 4 Conservative Dual Policy Optimization

When using nonlinear models, e.g. neural networks, the over-exploration issue causes unfavorable performance in practice, in terms of slow convergence and suboptimal asymptotic values. To tackle this challenge, the key is to abandon the sampling process and have guarantees *during* training.

In this regard, we propose *Conservative Dual Policy Optimization* (CDPO) that is simple yet provably efficient. By optimizing the policy following two successive update procedures iteratively, CDPO simultaneously enjoys monotonic policy value improvement and global optimality properties.

## 4.1 CDPO Framework

To begin with, consider the problem of maximizing the expected value, $\pi_t = \operatorname{argmax}_\pi \mathbb{E}[V_\pi^{f^*} | \mathcal{H}_t]$, where $\mathbb{E}[V^{f^*} | \mathcal{H}_t]$ denotes the expected values over the posterior. Obviously, we have the expected value improvement guarantee $\mathbb{E}[V_{\pi_t}^{f^*} | \mathcal{H}_t] \geq \mathbb{E}[V_{\pi_{t-1}}^{f^*} | \mathcal{H}_t]$. We can also perform expected value

maximization in a trust-region to guarantee iterative improvement under any $f^*$. However, such updates will lose the desired global convergence guarantee and may get stuck at local maxima even with linear models. For this reason, we propose a dual procedure of policy optimization.

**Referential Update.** The first update step returns an intermediate policy, denoted as $q_t$. This step is a greedy one in the sense that $q_t$ is optimal with respect to the value of a *single* model $\widetilde{f}_t$, which we call a *reference model*. Selecting a reference model and optimizing a policy w.r.t. it imitates the sampling-optimization procedure of PSRL. We will show in Section 5.1 that if we pose the constraint $\widetilde{f}_t \in \mathcal{F}_t$, then CDPO achieves the same expected regret as PSRL, which implies global optimality.

More importantly, policy optimization under $\widetilde{f}_t$ is more stable and can avoid the over-exploration issue in PSRL since we are free to set it as a steady reference between successive iterations. For example, we fix the reference model $\widetilde{f}_t$ as the least squares estimate $\widehat{f}_t^{LS}$ defined in (2.1), instead of a random model *sampled* from the large hypothesis that causes aggressive policy update. This gives us:

$$\text{Referential Update (with LS Reference):} \qquad q_t = \underset{q}{\operatorname{argmax}} V_q^{\widehat{f}_t^{LS}}. \tag{4.1}$$

**Constrained Conservative Update.** The conservative update then follows as the second stage of CDPO, which takes input $q_t$ and returns the reactive policy $\pi_{t+1}$:

$$\text{Conservative Update: } \pi_t = \underset{\pi}{\operatorname{argmax}} \mathbb{E}\left[V_\pi^{f_t} \mid \mathcal{H}_t\right], \text{ s.t. } \mathbb{E}_{s\sim\nu_{q_t}}\left[D_{\text{TV}}\big(\pi_t(\cdot|s), q_t(\cdot|s)\big)\right] \le \eta, \tag{4.2}$$

where $D_{\text{TV}}(\cdot, \cdot)$ stands for the total variation distance and $\eta$ is the hyperparameter that characterizes the trust-region constraint and controls the degree of exploration.

Compared with OFU-RL and PSRL, the above exploration and policy updates are conservative since the policy maximizes the *expectation* of the model value, instead of a *single* model's value (i.e. the optimistic model in OFU-RL and the sampled model in PSRL). The conservative update (4.2) avoids the pitfalls when the optimistic model or the posterior sampled model suffers large bias, which leads to aggressive policy updates and over-exploration during training. Notably, the term *conservative* in our work differs from previous use, e.g. Conservative Policy Iteration [23, 53]. While the latter refers to policy updates with constraints, ours is to emphasize the conservative range of randomness and the reduction of unnecessary over-exploration by shelving the sampling process.

In our analysis, we follow previous work [43, 59, 10, 35] and assume access to a policy optimization oracle. In practice, the problem of finding an optimal policy under a given model can be approximately solved by model-based solvers listed below. More fine-grained analysis can be obtained by applying off-the-shelf results established for policy gradient or MPC for specific policy or model function classes. This, however, is beyond the scope of this paper.

## 4.2 Practical Algorithm

The pseudocode of CDPO is in Alg. 1. The model-based solver $\texttt{MBPO}(\pi, f, \mathcal{J})$ outputs the policy ($q_t$ or $\pi_t$) that optimizes the objective $\mathcal{J}$ with access to model $f$. Several different types of solvers can be leveraged, e.g., model-augmented model-free policy optimization such as Dyna [61], model-based reparameterization gradient [18, 9], or model-predictive control [63]. Details of different optimization choices can be found in Appendix E. In experiments, we use Dyna and MPC solvers.

With Pinsker's inequality, the total variation constraint in (4.2) is replaced by the KL divergence [53, 2] in experiments. We follow previous work [34] to use neural network ensembles [10, 25] for model estimation and use calibrations [29, 10] for accurate uncertainty measure.

---

**Algorithm 1** Practical CDPO Algorithm

---

**Input:** Prior $\phi$, model-based policy optimization solver $\texttt{MBPO}(\pi, f, \mathcal{J})$.

1: **for** iteration $t = 1, ..., T$ **do**
2:    $q_t \leftarrow \texttt{MBPO}(\cdot, \widehat{f}_t^{LS}, (4.1))$
3:    Sample $N$ models $\{f_{t,n}\}_{n=1}^N$
4:    $\pi_t \leftarrow \texttt{MBPO}(q_t, \{f_{t,n}\}_{n=1}^N, (4.2))$
5:    Execute $\pi_t$ in the real MDP
6:    Update $\mathcal{H}_{t+1} = \mathcal{H}_t \cup \{s_{h,t}, a_{h,t}, s_{h+1,t}\}_h$
7:    Update $\widehat{f}_{t+1}^{LS}$ and $\phi$
8: **end for**
9: **return** policy $\pi_T$

---

# 5 Analysis

In this section, we first show the statistical equivalence between CDPO and PSRL in terms of the same BayesRegret bound. Then we give the iterative policy value bound with monotonic improvement. Finally, we prove the global convergence of CDPO. The missing proofs can be found in the Appendix.

## 5.1 Statistical Equivalence between CDPO and PSRL

We begin our analysis by highlighting the connection between CDPO and PSRL with the following theorem, from which we also show the role of the dual update procedure and the reference model.

**Theorem 5.1** (CDPO Matches PSRL in BayesRegret)**.** Let $\pi^{\text{PSRL}}$ be the policy of any posterior sampling algorithm for reinforcement learning optimized by (3.2). If the BayesRegret bound of $\pi^{\text{PSRL}}$ satisfies that for any $T > 0$, $\text{BayesRegret}(T, \pi^{\text{PSRL}}, \phi) \leq \mathcal{D}$, then for all $T > 0$, we have for the CDPO policy $\pi^{\text{CDPO}}$ that $\text{BayesRegret}(T, \pi^{\text{CDPO}}, \phi) \leq 3\mathcal{D}$.

*Sketch proof.* We first sketch the general strategy in the PSRL analysis. Recall the definition of the Bayesian expected regret $\text{BayesRegret}(T, \pi, \phi) := \mathbb{E}[\sum_{t=1}^{T} \mathfrak{R}_t]$, where $\mathfrak{R}_t = V_{\pi^*}^{f^*} - V_{\pi_t}^{f^*}$. PSRL breaks down $\mathfrak{R}_t$ by adding and subtracting $V_{\pi_{f_t}}^{f_t}$, the value of the *imagined* optimal policy $\pi_{f_t}$ under a sampled model $f_t$, i.e. $\pi_{f_t} = \text{argmax}_{\pi} V_{\pi}^{f_t}$.

$$\text{PSRL:} \qquad \mathfrak{R}_t = V_{\pi^*}^{f^*} - V_{\pi_t}^{f^*} = V_{\pi^*}^{f^*} - V_{\pi_{f_t}}^{f^*} = V_{\pi^*}^{f^*} - V_{\pi_{f_t}}^{f_t} + V_{\pi_{f_t}}^{f_t} - V_{\pi_{f_t}}^{f^*}, \qquad (5.1)$$

where the second equality follows from the definition of the PSRL policy. Following the law of total expectation and the Posterior Sampling Lemma (e.g. Lemma 1 in [42]), we have $\mathbb{E}[V_{\pi^*}^{f^*} - V_{\pi_{f_t}}^{f_t}] = 0$ by noting that $f^*$ and $f_t$ are identically distributed conditioned upon $\mathcal{H}_t$. Then we obtain

$$\text{BayesRegret}(T, \pi^{\text{PSRL}}, \phi) = \sum_{t=1}^{T} \mathbb{E}[V_{\pi_{f_t}}^{f_t} - V_{\pi_{f_t}}^{f^*}] \leq \gamma \sum_{t=1}^{T} \mathbb{E}\Big[\mathbb{E}_{\rho}\big[L\|f_t(s_h, a_h) - f^*(s_h, a_h)\|_2\big]\Big]$$

$$\leq \gamma \frac{L}{1 - 4\delta} \sum_{t=1}^{T} \mathbb{E}[\omega_t] + 4\gamma\delta T \leq \mathcal{D}, \qquad (5.2)$$

where the first inequality follows from the simulation lemma under the $L$-Lipschitz value assumption [43]. The second inequality follows from the definition of $\omega_t$ in (2.6) and the construction of confidence set such that $\mathbb{P}(f^* \in \bigcap \mathcal{F}_t) \geq 1 - 2\delta$ and $\mathbb{P}(f_t \in \bigcap \mathcal{F}_t, f^* \in \bigcap \mathcal{F}_t) \geq 1 - 4\delta$ via a union bound. As more data is collected, the model uncertainty is reduced and the sum of confidence set width $\omega_t$ will be sublinear in $T$ (c.f. Lemma B.5 and B.6), indicating sublinear regret.

When it comes to CDPO, we decompose the regret as

$$\text{CDPO:} \qquad \mathfrak{R}_t = V_{\pi^*}^{f^*} - V_{\pi_t}^{f^*} = V_{\pi^*}^{f^*} - V_{\pi_{f_t}}^{f_t} + V_{\pi_{f_t}}^{f_t} - V_{\pi_t}^{f_t} + V_{\pi_t}^{f_t} - V_{\pi_t}^{f^*}, \qquad (5.3)$$

where the CDPO policy $\pi_t$ is defined in (4.2). Since $\mathbb{E}[V_{\pi^*}^{f^*} - V_{\pi_{f_t}}^{f_t}] = 0$, we have

$$\text{BayesRegret}(T, \pi^{\text{CDPO}}, \phi) = \sum_{t=1}^{T} \mathbb{E}[V_{\pi_{f_t}}^{f_t} - V_{\pi_t}^{f_t} + V_{\pi_t}^{f_t} - V_{\pi_t}^{f^*}]$$

$$\leq \sum_{t=1}^{T} \mathbb{E}[V_{\pi_{f_t}}^{f_t} - V_{\pi_{f_t}}^{\widetilde{f}_t} + V_{q_t}^{\widetilde{f}_t} - V_{q_t}^{f_t} + V_{\pi_t}^{f_t} - V_{\pi_t}^{f^*}] \leq \frac{L}{1 - 4\delta} \sum_{t=1}^{T} 3\mathbb{E}[\omega_t] + 8\gamma\delta T \leq 3\mathcal{D}, \ (5.4)$$

where the first inequality follows from the greediness of $q_t$ and $\pi_t$ in the dual update steps, i.e., $V_{\pi_{f_t}}^{\widetilde{f}_t} \leq V_{q_t}^{\widetilde{f}_t}$ for any $\pi_{f_t}$ as well as $\mathbb{E}[V_{\pi_t}^{f_t}] \geq \mathbb{E}[V_{q_t}^{f_t}]$. The $8\delta T$ term is introduced since $\widetilde{f}_t \in \mathcal{F}_t$ and $\mathbb{P}(f_t \in \bigcap \mathcal{F}_t, \widetilde{f}_t \in \bigcap \mathcal{F}_t) \geq 1 - 2\delta$. $\qquad \square$

Theorem 5.1 indicates that although CDPO performs conservative updates and abandons the sampling process, it matches the statistical efficiency of PSRL up to constant factors.

The importance of the reference model and the dual procedure is also reflected in the proof. The referential update builds the bridge between $V_{\pi_{f_t}}^{f_t}$ and $V_{\pi_t}^{f_t}$. Policy optimization under the reference model mimics the sampling-then-optimization procedure of PSRL while offering more stability when the reference is steady, e.g., the least squares estimate we use. We formalize this idea below.

## 5.2 CDPO Policy Iterative Improvement

One motivation for the conservative update is that it maximizes (thus improves) the expected value over the posterior. In this section, we are interested in the policy value improvement under any unknown $f^*$. Namely, we seek to have the iterative improvement bound $J(\pi_t) - J(\pi_{t-1})$, where the true objective $J$ is defined in (2.3).

We impose the following regularity conditions on the underlying MDP transition and the state-action visitation.

**Assumption 5.2** (Regularity Condition on MDP Transition). Assume that the MDP transition function $f^* : \mathcal{S} \times \mathcal{A} \to \mathcal{S}$ is with additive $\sigma$-sub-Gaussian noise and bounded norm, i.e., $\|s\|_2 \leq C$.

**Assumption 5.3** (Regularity Condition on State-Action Visitation). We assume that there exists $\kappa > 0$ such that for any policy $\pi_t$, $t \in [1, T]$,

$$\left\{ \mathbb{E}_{\rho_{\pi_t}} \left[ \left( \frac{d\rho_{q_{t+1}}}{d\rho_{\pi_t}}(s, a) \right)^2 \right] \right\}^{1/2} \leq \kappa, \tag{5.5}$$

where $d\rho_{q_{t+1}}/d\rho_{\pi_t}$ is the Radon-Nikodym derivative of $\rho_{q_{t+1}}$ with respect to $\rho_{\pi_t}$.

**Theorem 5.4** (Policy Iterative Improvement). Suppose we have $\|\widetilde{f}(\cdot, \cdot)\| \leq C$ for $\widetilde{f} \in \mathcal{F}$ where the model class $\mathcal{F}$ is finite. Define $\iota := \max_{s,a} |A_\pi^{f^*}(s, a)|$, where $A_\pi^{f^*}$ is the advantage function defined as $A_\pi^{f^*}(s, a) := Q_\pi^{f^*}(s, a) - V_\pi^{f^*}(s)$. With probability at least $1 - \delta$, the policy improvement between successive iterations is bounded by

$$J(\pi_t) - J(\pi_{t-1}) \geq \Delta(t) - (1 + \kappa) \cdot \frac{22\gamma C^2 \ln(|\mathcal{F}|/\delta)}{(1-\gamma)H} - \frac{2\eta\iota}{1-\gamma}, \tag{5.6}$$

where $\Delta(t) := \mathbb{E}_{s \sim \zeta} \left[ V_{q_t}^{\widetilde{f}_t}(s) - V_{q_{t-1}}^{\widetilde{f}_t}(s) \right] \geq 0$ due to the greediness of $q_t$.

The above theorem provides the iterative improvement bound following the CDPO algorithm. When $H$ is large enough, the policy value improvement is at least $\Delta(t)$ by choosing a properly small $\eta$.

In particular, the first term $\Delta(t)$ characterizes the policy improvement brought by the greedy exploitation in (4.1), and $\Delta(t) \geq 0$ since $q_t$ is optimal under the reference model $\widetilde{f}_t$. The second term in (5.6) accounts for the generalization error of least square methods. Specifically, model $\widetilde{f}_t = \widehat{f}_t^{LS} \in \mathcal{F}_t$ is trained to fit the history samples. However, we seek to have the model error bound over the state-action visitation measure, which requires the deviation from the empirical mean to its expectation using Bernstein's inequality and union bound. Finally, the trust-region constraint in (4.2) brings the $4\eta\alpha/(1-\gamma)$ term, which reduces to zero if $\eta$ is small. This makes intuitive sense as $\eta$ controls the degree of conservative exploration.

## 5.3 Global Optimality of CDPO

We now analyze the global optimality of CDPO by studying its expected regret. As discussed in Section 3, agnostic reinforcement learning is impossible. Without structural assumptions, additional complexity measure is required for a generalization bound beyond finite settings. For this reason, we adopt the notation of *eluder dimension* [49, 43], defined as follows:

**Definition 5.5** (($\mathcal{F}, \varepsilon$)-Dependence). If we say $(s, a) \in \mathcal{S} \times \mathcal{A}$ is ($\mathcal{F}, \varepsilon$)-dependent on $\{(s_i, a_i)\}_{i=1}^n \subseteq \mathcal{S} \times \mathcal{A}$, then

$$\forall f_1, f_2 \in \mathcal{F}, \sum_{i=1}^n \|f_1(s_i, a_i) - f_2(s_i, a_i)\|_2^2 \leq \varepsilon^2 \Rightarrow \|f_1(s, a) - f_2(s, a)\|_2 \leq \varepsilon.$$

Conversely, $(s, a) \in \mathcal{S} \times \mathcal{A}$ is ($\mathcal{F}, \varepsilon$)-independent of $\{(s_i, a_i)\}_{i=1}^n$ if and only if it does not satisfy the definition for dependence.

**Definition 5.6** (Eluder Dimension). The eluder dimension $dim_E(\mathcal{F}, \varepsilon)$ is the length of the longest possible sequence of elements in $\mathcal{S} \times \mathcal{A}$ such that for some $\varepsilon' \geq \varepsilon$, every element is ($\mathcal{F}, \varepsilon'$)-independent of its predecessors.

We make the following assumption on the Lipschitz continuity of the value function.

**Assumption 5.7** (Lipschitz Continuous Value). At iteration $t$, assume the value function $V_\pi^{f_t}$ for any policy $\pi$ is Lipschitz continuous in the sense that $|V_\pi^{f_t}(s_1) - V_\pi^{f_t}(s_2)| \leq L_t \|s_1 - s_2\|_2$.

Notably, Assumption 5.7 holds under certain regularity conditions of the MDP, e.g. when the transition and rewards are Lipschitz continuous [5, 47]. Under this assumption, many RL settings can be satisfied [13], e.g., nonlinear models with stochastic Lipschitz policies and Lipschitz reward models, and is thus adopted by various model-based RL work [35, 7, 13].

We now study the global optimality of CDPO by the following expected regret theorem, which can be seen as a direct consequence of Theorem 5.1 that states the statistical equivalence between CDPO and PSRL.

**Theorem 5.8** (Expected Regret of CDPO). Let $N(\mathcal{F}, \alpha, \|\cdot\|_2)$ be the $\alpha$-covering number of $\mathcal{F}$. Denote $d_E := \dim_E(\mathcal{F}, T^{-1})$ for the eluder dimension of $\mathcal{F}$ at precision $1/T$. Under Assumption 5.2 and 5.7, the cumulative expected regret of CDPO in $T$ iterations is bounded by

$$\text{BayesRegret}(T, \pi, \phi) \leq \frac{\gamma T(3T-5)L}{(T-1)(T-2)} \cdot \left(1 + \frac{1}{1-\gamma} Cd_E + 4\sqrt{Td_E\beta}\right) + 4\gamma C, \tag{5.7}$$

where $\beta := 8\sigma^2 \log\left(2N\left(\mathcal{F}, 1/(T^2), \|\cdot\|_2\right)T\right) + 2\left(8C + \sqrt{8\sigma^2 \log(8T^3)}\right)/T$ and $L := \mathbb{E}[L_t]$.

Here, the covering number is introduced since we are considering $\mathcal{F}$ that may contain infinitely many functions, for which we cannot simply apply a union bound. Besides, $\beta$ is the confidence parameter that contains $f^*$ with high probability (via concentration inequality).

To clarify the asymptotics of the expected regret bound, we introduce another measure of dimensionality that captures the sensitivity of $\mathcal{F}$ to statistical overfitting.

**Corollary 5.9** (Asymptotic Bound). Define the Kolmogorov dimension w.r.t. function class $\mathcal{F}$ as

$$d_K = \dim_K(\mathcal{F}) := \limsup_{\alpha \downarrow 0} \frac{\log(N(\mathcal{F}, \alpha, \|\cdot\|_2))}{\log(1/\alpha)}.$$

Under the assumptions of Theorem 5.8 and by omitting terms logarithmic in $T$, the regret of CDPO is

$$\text{BayesRegret}(T, \pi, \phi) = \widetilde{O}(L\sigma\sqrt{d_K d_E T}). \tag{5.8}$$

The sublinear regret result permits the global optimality and sample efficiency for any model class with a reasonable complexity measure. Meanwhile, the iterative improvement theorem guarantees efficient exploration and good performance even when the model class is highly nonlinear.

## 6 Empirical Evaluation

### 6.1 Understanding Different Exploration Mechanisms

We first provide insights and evidence of why CDPO exploration can be more efficient in the tabular $N$-Chain MDPs, which have optimal *right* actions and suboptimal *left* actions at each of the $N$ states. Settings and full results are provided in Appendix F.2. In Figure 1, we compare the posterior of CDPO and PSRL at the state that is the furthest away from the initial state, i.e. the state that is the hardest for the agents to reach and explore.

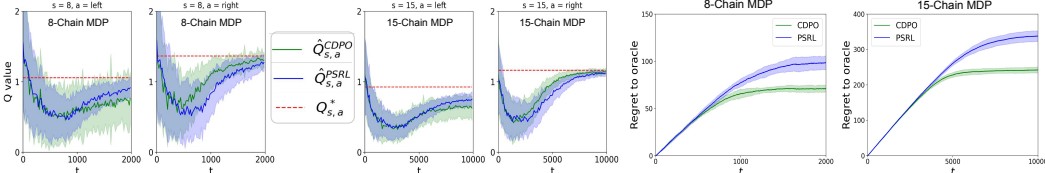

Figure 1: CDPO and PSRL posterior on an 8-Chain MDP and a 15-Chain MDP, where the *right* actions are optimal.

Figure 2: Regret curve of CDPO and PSRL when $N = 8$ and $N = 15$.

When training starts, both algorithms have a large variance of value estimation. However, as training progresses, CDPO gives more accurate and certain estimates, but *only* for the optimal *right* actions not

for the suboptimal *left* actions, while PSRL agents explore *both* directions. This verifies the potential over-exploration issue in PSRL: as long as the uncertainty contains unrealistically large values, PSRL agents can perform uninformative exploration by acting suboptimally according to an inaccurate *sampled* model. In contrast, CDPO replaces the sampled model with a stable mean estimate and cares about the *expected* value, thus avoiding such pitfalls. We see in Figure 2 that although CDPO has much larger uncertainty for the suboptimal *left* actions, its regret is lower.

## 6.2 Exploration Efficiency with Nonlinear Model Class

In finite MDPs, PSRL-style agents can specify and try every possible action to finally obtain an accurate high-confidence prediction. However, our discussion in Section 3 indicates that a similar over-exploration issue in more complex environments can lead to less informative exploration steps, which only eliminate an exponentially small portion of the uncertainty.

To see its impact on the training performance, we report the results of provable algorithms with nonlinear models on several MuJoCo tasks in Figure 3. For OFU-RL, we mainly evaluate HUCRL [10], a deep algorithm proposed to deal with the intractability of the joint optimization. We observe that all algorithms achieve asymptotic optimality in the inverted pendulum. Since the dimension of the pendulum task is low, learning an accurate (and thus generalizable) model poses no actual challenge. However, in higher dimensional tasks such as half-cheetah, CDPO achieves a higher asymptotic value with faster convergence. Implementation details and hyperparameters are provided in Appendix F.1.

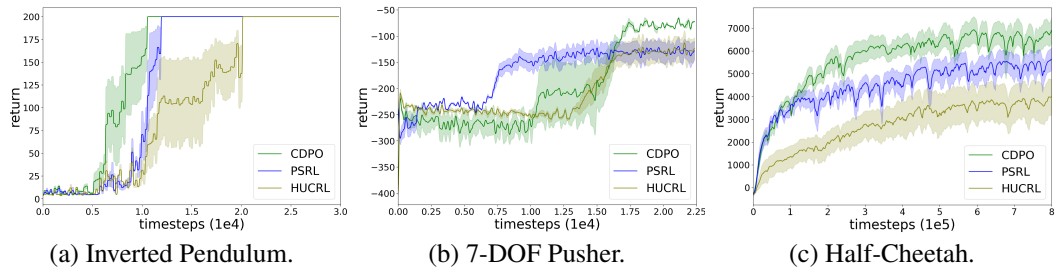

(a) Inverted Pendulum.      (b) 7-DOF Pusher.      (c) Half-Cheetah.

Figure 3: Performance of CDPO, PSRL, and HUCRL equipped with nonlinear models in several MuJoCo tasks: inverted pendulum swing-up, pusher goal-reaching, and half-cheetah locomotion.

## 6.3 Comparison with Prior RL Algorithms

We also examine a broader range of MBRL algorithms, including MBPO [20], SLBO [35], and ME-TRPO [30]. The model-free baselines include SAC [16], PPO [54], and MPO [2]. The results are shown in Figure 4. We observe that CDPO achieves competitive or higher asymptotic performance while requiring fewer samples compared to both the model-based and the model-free baselines.

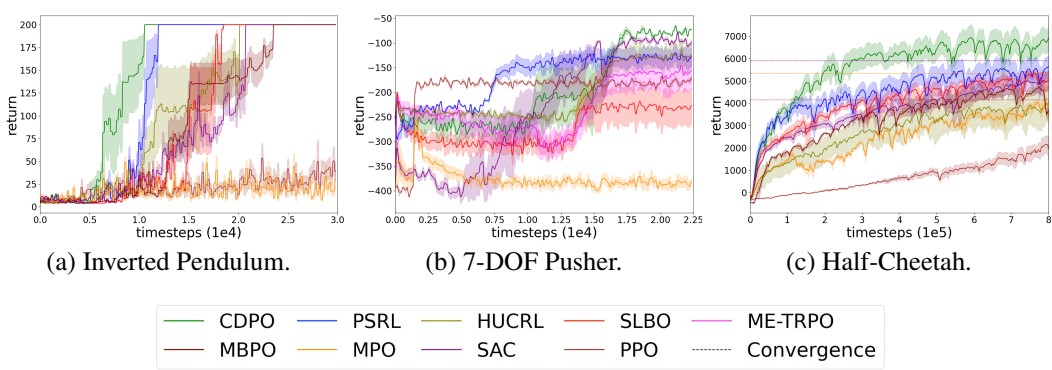

(a) Inverted Pendulum.      (b) 7-DOF Pusher.      (c) Half-Cheetah.

Figure 4: Comparison between CDPO and model-free, model-based RL baseline algorithms.

## 6.4 Ablation Study

We conduct ablation studies to provide a better understanding of the components in CDPO. One can observe from Figure 5 that the policies updated with only Referential Update or Conservative Update lag behind the dual framework. We also test the necessity and sensitivity of the constraint hyperparameter $\eta$. We see that a constant $\eta$ and a time-decayed $\eta$ achieve similar asymptotic values with a similar convergence rate, showing the robustness of CDPO. However, removing the constraint will lose the policy improvement guarantee, thus causing degradation. Ablation on different choices of `MBPO` solver (Dyna and POPLIN-P [63]) shows the generalizability of CDPO.

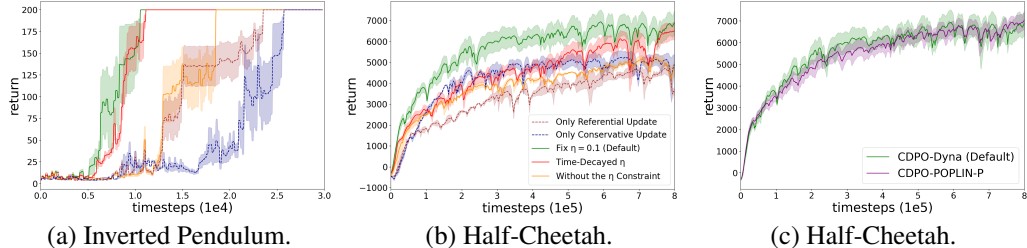

| (a) Inverted Pendulum. | (b) Half-Cheetah. | (c) Half-Cheetah. |

Figure 5: Ablation studies on the effect of the dual update steps and the trust-region constraint. The robustness and generalizability of the CDPO framework are demonstrated by the results of different choices of the constraint threshold and different solvers.

## 7    Conclusions & Future Work

In this work, we present *Conservative Dual Policy Optimization* (CDPO), a simple yet provable model-based algorithm. By iterative execution of the *Referential Update* and *Conservative Update*, CDPO explores within a reasonable range while avoiding aggressive policy update. Moreover, CDPO gets rid of the harmful sampling procedure in previous provable approaches. Instead, an intermediate policy is optimized under a stable *reference* model, and the agent conservatively explore the environment by maximizing the *expected* policy value. With the same order of regret as PSRL, the proposed algorithm can achieve global optimality while monotonically improving the policy. Considering our naive choice of the reference model, other more sophisticated designs should be a fruitful future direction. It will also be interesting to explore different choices of the `MBPO` solvers, which we would like to leave as future work.

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
