# A  Proofs

## A.1  Proof of Theorem 5.4

*Proof.* We lay out the proof in two major steps. Firstly, we characterize the performance difference between $J(q_t)$ and $J(\pi_{t-1})$, which can be done by applying Lemma B.3. Specifically, we set $\pi_1$, $\pi_2$ in Lemma B.3 to $q_t$, $\pi_{t-1}$ and set $f$ as the reference model $\widetilde{f}_t$. Then we obtain

$$
\begin{aligned}
&J(q_t) - J(\pi_{t-1}) \\
&= \Delta(t) - \frac{\gamma}{2(1-\gamma)}\left( \mathbb{E}_{\rho_{q_t}}\left[ \left\| \widetilde{f}_t(s,a) - f^*(\cdot|s,a) \right\|_1 \right] + \mathbb{E}_{\rho_{\pi_{t-1}}}\left[ \left\| \widetilde{f}_t(s,a) - f^*(\cdot|s,a) \right\|_1 \right] \right),
\end{aligned}
\tag{A.1}
$$

where $\Delta(t) := \mathbb{E}_{s\sim\zeta}\left[ V_{q_t}^{\widetilde{f}_t}(s) - V_{\pi_{t-1}}^{\widetilde{f}_t}(s) \right] \geq 0$ due to the optimality of $q_t$ under $\widetilde{f}_t$, i.e., $q_t = \mathrm{argmax}_q V_q^{\widetilde{f}_t}$.

Recall that the reference model is the least squares estimate, i.e.,

$$
\widetilde{f}_t = \widehat{f}_t^{LS} = \underset{f\in\mathcal{F}}{\mathrm{argmin}} \sum_{(s,a,s')\in\mathcal{H}_{t-1}} \left\| f(s,a) - s' \right\|_2^2,
$$

where $\mathcal{H}_{t-1}$ is the trajectory in the real environment when following policy $\pi_{t-1}$.

From the simulation property of continuous distribution, we have the following equivalence between the direct and indirect ways of drawing samples:

$$
s' \sim f^*(\cdot|s,a) \;\equiv\; s' = f^*(s,a) + \epsilon, \epsilon \sim p(\epsilon),
$$

where $p(\epsilon)$ is some noise distribution.

Therefore, according to the Gaussian noise assumption, we obtain from the least squares generalization bound in Lemma B.4 that

$$
\mathbb{E}_{\rho_{\pi_{t-1}}}\left[ \left\| \widetilde{f}_t(s,a) - f^*(\cdot|s,a) \right\|_1 \right] \leq \frac{22C^2 \ln(|\mathcal{F}|/\delta)}{H},
\tag{A.2}
$$

where $\epsilon_{\mathrm{approx}} = 0$ in the generalization bound as the realizability is guaranteed since $\widehat{f}_t^{LS}$ and $f^*$ are from the same function class $\mathcal{F}$.

Similarly, we have for the intermediate policy $q_t$ that

$$
\mathbb{E}_{\rho_{q_t}}\left[ \left\| \widetilde{f}_t(s,a) - f^*(\cdot|s,a) \right\|_1 \right] \leq \mathbb{E}_{\rho_{\pi_{t-1}}}\left[ \left\| \widetilde{f}_t(s,a) - f^*(\cdot|s,a) \right\|_1 \right] \cdot \left\{ \mathbb{E}_{\rho_{\pi_{t-1}}}\left[ \left( \frac{d\rho_{q_t}}{d\rho_{\pi_{t-1}}}(s) \right)^2 \right] \right\}^{1/2}
$$

$$
\leq \kappa \cdot \frac{22C^2 \ln(|\mathcal{F}|/\delta)}{H}.
\tag{A.3}
$$

Now we can bound (A.1) by

$$
J(q_t) - J(\pi_{t-1}) \geq \Delta(t) - (1+\kappa) \cdot \frac{22\gamma C^2 \ln(|\mathcal{F}|/\delta)}{(1-\gamma)H}.
\tag{A.4}
$$

The second step of the proof is to characterize the performance difference between $J(\pi_t)$ and $J(q_t)$. From the Performance Difference Lemma B.2, we obtain

$$
\begin{aligned}
J(q_t) - J(\pi_t) &= \frac{1}{1-\gamma} \cdot \mathbb{E}_{(s,a)\sim\rho_{q_t}}\left[ A_{\pi_t}^{f^*}(s,a) \right] \\
&= \frac{1}{1-\gamma} \cdot \mathbb{E}_{s\sim\nu_{q_t}}\left[ \mathbb{E}_{a\sim q_t}\left[ A_{\pi_t}^{f^*}(s,a) \right] \right] \\
&= \frac{1}{1-\gamma} \cdot \mathbb{E}_{s\sim\nu_{q_t}}\left[ \mathbb{E}_{a\sim q_t}\left[ A_{\pi_t}^{f^*}(s,a) \right] - \mathbb{E}_{a\sim\pi_t}\left[ A_{\pi_t}^{f^*}(s,a) \right] \right],
\end{aligned}
\tag{A.5}
$$

where recall that $\iota := \max_{s,a} |A_\pi^{f^*}(s,a)|$ and the third equality holds due to $\mathbb{E}_{a\sim\pi_t}\left[A_{\pi_t}^{f^*}(s,a)\right] = 0$ for any $s$.

By the definition of the total variation distance, we can further bound the absolute difference as

$$|J(q_t) - J(\pi_t)| \leq \frac{2\eta\iota}{1-\gamma}, \tag{A.6}$$

Thus, we have $J(\pi_t) - J(q_t) \geq -2\eta\iota/(1-\gamma)$ and similarly $J(q_{t-1}) - J(\pi_{t-1}) \geq -2\eta\iota/(1-\gamma)$. Combining with (A.4) gives us the iterative improvement bound as follows:

$$
\begin{aligned}
J(\pi_t) - J(\pi_{t-1}) &= J(\pi_t) - J(q_t) + J(q_t) - J(\pi_{t-1}) \\
&\geq \Delta(t) - (1+\kappa)\cdot \frac{22\gamma C^2 \ln(|\mathcal{F}|/\delta)}{(1-\gamma)H} - \frac{2\eta\iota}{1-\gamma}. 
\end{aligned} \tag{A.7}
$$

$\square$

## A.2   Proof of Theorem 5.8

*Proof.* We are interested in the expected regret defined as $\text{BayesRegret}(T,\pi,\phi) := \mathbb{E}[\sum_{t=1}^T \mathfrak{R}_t]$, where $\mathfrak{R}_t = V_{\pi^*}^{f^*} - V_{\pi_t}^{f^*}$.

Recall the definition of the reactive policy $\pi_t$ in CDPO (i.e. (4.2)) and the imagined best-performing policy $\pi_{f_t}$ under a sampled model $f_t$, i.e., $\pi_{f_t} = \max_\pi V_\pi^{f_t}$.

From the Posterior Sampling Lemma, we know that if $\psi$ is the distribution of $f^*$, then for any sigma-algebra $\sigma(\mathcal{H}_t)$-measurable function $g$,

$$\mathbb{E}[g(f^*)\,|\,\mathcal{H}_t] = \mathbb{E}[g(f_t)\,|\,\mathcal{H}_t]. \tag{A.8}$$

The PS Lemma together with the law of total expectation gives us

$$\mathbb{E}[V_{\pi^*}^{f^*} - V_{\pi_{f_t}}^{f_t}] = 0, \tag{A.9}$$

where the equality holds since the true $f^*$ and the sampled $f_t$ are identically distributed when conditioned on $\mathcal{H}_t$. Therefore, we obtain the expected regret for CDPO as

$$
\begin{aligned}
\text{BayesRegret}(T,\pi,\phi) &= \sum_{t=1}^T \mathbb{E}[V_{\pi_{f_t}}^{f_t} - V_{\pi_t}^{f_t} + V_{\pi_t}^{f_t} - V_{\pi_t}^{f^*}] \\
&= \sum_{t=1}^T \mathbb{E}[V_{\pi_{f_t}}^{f_t} - V_{\pi_{f_t}}^{\widetilde{f}_t} + V_{\pi_{f_t}}^{\widetilde{f}_t} - V_{\pi_t}^{f_t} + V_{\pi_t}^{f_t} - V_{\pi_t}^{f^*}] \\
&\leq \sum_{t=1}^T \mathbb{E}[V_{\pi_{f_t}}^{f_t} - V_{\pi_{f_t}}^{\widetilde{f}_t} + V_{q_t}^{\widetilde{f}_t} - V_{q_t}^{f_t} + V_{\pi_t}^{f_t} - V_{\pi_t}^{f^*}], \tag{A.10}
\end{aligned}
$$

where the inequality follows from the greediness of $q_t$ and the optimality of $\pi_t$ within a trust-region centered around $q_t$, i.e., $V_{\pi_{f_t}}^{\widetilde{f}_t} \leq V_{q_t}^{\widetilde{f}_t}$ for any $\pi_{f_t}$ and $V_{\pi_t}^{f_t} \geq V_{q_t}^{f_t}$.

From the Simulation Lemma B.1, we have the bound of $\mathbb{E}\left[\left|V_\pi^{f_t} - V_\pi^{\widetilde{f}_t}\right|\right]$ for any policy $\pi$ as follows:

$$
\begin{aligned}
\mathbb{E}\left[\left|V_\pi^{f_t} - V_\pi^{\widetilde{f}_t}\right|\right] &= \gamma\mathbb{E}\left[\left|\mathbb{E}_{(s,a)\sim\widetilde{\rho}_\pi}[(f_t(\cdot|s,a) - \widetilde{f}_t(\cdot|s,a))\cdot V_\pi^{f_t}(s,a)]\right|\right] \\
&\leq \gamma\mathbb{E}\left[\left|\mathbb{E}_{(s,a)\sim\widetilde{\rho}_\pi}[L_t\cdot\|f_t(s,a) - \widetilde{f}_t(s,a)\|_2]\right|\right], \tag{A.11}
\end{aligned}
$$

where the first equation follows from Lemma B.1 and $\widetilde{\rho}_\pi$ is the state-action visitation measure under model $\widetilde{f}_t$, the second inequality follows the simulation property of continuous distribution and the Lipschitz value function assumption.

We define the event $A = \left\{ \widetilde{f}_t \in \bigcap_t \mathcal{F}_t, f_t \in \bigcap_t \mathcal{F}_t \right\}$. Recall that the model is bounded by $\|f\|_2 \leq C$. Then we can reduce the expected regret to a sum of set widths:

$$\mathbb{E}\left[V_\pi^{f_t} - V_\pi^{\widetilde{f}_t}\right] \leq \gamma \mathbb{E}\left[\left|\mathbb{E}_{(s,a)\sim\widetilde{\rho}_\pi}\left[\mathbb{E}[L_t|A]\omega_t(s,a) + \left(1 - \mathbb{P}(A)\right)C\right]\right|\right]. \tag{A.12}$$

We can further know from the construction of the confidence set (c.f. Lemma B.5) that $\mathbb{P}\left(f^* \in \bigcap_t \mathcal{F}_t\right) \geq 1 - 2\delta$ and $\mathbb{P}(A) \geq 1 - 2\delta$ since $f_t, f^*$ are identically distributed and $\mathbb{P}\left(\widetilde{f}_t \in \mathcal{F}_t\right) = 1$ as $\mathcal{F}_t$ is centered at the least squares model for all $t$.

Besides, we have for

$$\mathbb{E}[L_t|A] \leq \frac{L_t}{P(A)} \leq \frac{L_t}{1 - 2\delta}. \tag{A.13}$$

Plugging into (A.21), we have

$$\mathbb{E}\left[V_\pi^{f_t} - V_\pi^{\widetilde{f}_t}\right] \leq \gamma \mathbb{E}\left[\left|\mathbb{E}_{(s,a)\sim\widetilde{\rho}_\pi}\left[L_t/(1 - 2\delta)\omega_t(s,a) + 2\delta C\right]\right|\right]$$

$$\leq \gamma \mathbb{E}\left[\frac{L_t}{1 - 2\delta} \cdot \left|\mathbb{E}_{(s,a)\sim\widetilde{\rho}_\pi}\left[\omega_t(s,a)\right]\right|\right] + 2\gamma\delta C. \tag{A.14}$$

Summing over $T$ iterations gives us

$$\sum_{t=1}^T \mathbb{E}\left[V_\pi^{f_t} - V_\pi^{\widetilde{f}_t}\right] \leq \gamma \sum_{t=1}^T \mathbb{E}\left[\frac{L_t}{1 - 2\delta} \cdot \left|\mathbb{E}_{(s,a)\sim\widetilde{\rho}_\pi}\left[\omega_t(s,a)\right]\right|\right] + 2\gamma\delta CT. \tag{A.15}$$

By setting $\delta = 1/(2T)$, we obtain

$$\sum_{t=1}^T \mathbb{E}\left[V_\pi^{f_t} - V_\pi^{\widetilde{f}_t}\right] \leq \frac{\gamma LT}{T - 1} \sum_{t=1}^T \mathbb{E}_{\widetilde{\rho}_\pi}[\omega_t(s,a)] + \gamma C$$

$$\leq \frac{\gamma LT}{T - 1} \cdot \left(1 + \frac{1}{1 - \gamma}Cd_E + 4\sqrt{Td_E\beta_T(1/(2T),\alpha)}\right) + \gamma C, \tag{A.16}$$

where the last inequality follows from Lemma B.6 to bound the sum of the set width. We denote $d_E := \dim_E(\mathcal{F}, T^{-1})$ for notation simplicity.

Since (A.16) holds for all policy $\pi$, we have the bound for $\mathbb{E}[V_{\pi_{f_t}}^{f_t} - V_{\pi_{f_t}}^{\widetilde{f}_t}]$ and the bound for $\mathbb{E}[V_{q_t}^{\widetilde{f}_t} - V_{q_t}^{f_t}]$. What remains in the expected regret (A.19) is the $\mathbb{E}[V_{\pi_t}^{f_t} - V_{\pi_t}^{f^*}]$ term, which can be bounded similarly.

Specifically, we define another event $B = \left\{ f^* \in \bigcap_t \mathcal{F}_t, f_t \in \bigcap_t \mathcal{F}_t \right\}$. Since by construction $\mathbb{P}\left(f^* \in \bigcap_t \mathcal{F}_t\right) \geq 1 - 2\delta$ and $\mathbb{P}\left(f_t \in \bigcap_t \mathcal{F}_t\right) \geq 1 - 2\delta$, we have $\mathbb{P}(B) \geq 1 - 4\delta$ via a union bound. This implies the following bound

$$\sum_{t=1}^T \mathbb{E}\left[V_\pi^{f_t} - V_\pi^{f^*}\right] \leq \gamma \sum_{t=1}^T \mathbb{E}\left[\frac{L_t}{1 - 4\delta} \cdot \left|\mathbb{E}\left[\omega_t(s,a)\right]\right|\right] + 4\gamma\delta CT$$

$$\leq \frac{\gamma LT}{T - 2} \sum_{t=1}^T \mathbb{E}_{\rho_\pi}[\omega_t(s,a)] + 2\gamma C$$

$$\leq \frac{\gamma LT}{T - 2} \cdot \left(1 + \frac{1}{1 - \gamma}Cd_E + 4\sqrt{Td_E\beta_T(1/(2T),\alpha)}\right) + 2\gamma C, \tag{A.17}$$

where the second inequality follows from the choice of $\delta$, i.e., $\delta = 1/(2T)$.

Plugging (A.16) and (A.17) into (A.19), we obtain the expected regret as

$$
\begin{aligned}
\text{BayesRegret}(T, \pi, \phi) &\leq \sum_{t=1}^{T} \mathbb{E}[V_{\pi_{f_t}}^{f_t} - V_{\pi_{f_t}}^{\widetilde{f}_t} + V_{q_t}^{\widetilde{f}_t} - V_{q_t}^{f_t} + V_{\pi_t}^{f_t} - V_{\pi_t}^{f^*}] \\
&\leq \left( \frac{2\gamma LT}{T-1} + \frac{\gamma LT}{T-2} \right) \cdot \left( 1 + \frac{1}{1-\gamma} C d_E + 4\sqrt{T d_E \beta_T(1/(2T), \alpha)} \right) + 4\gamma C \\
&= \frac{\gamma T(3T-5)L}{(T-1)(T-2)} \cdot \left( 1 + \frac{1}{1-\gamma} C d_E + 4\sqrt{T d_E \beta_T(1/(2T), \alpha)} \right) + 4\gamma C.
\end{aligned}
$$
(A.18)

By setting $\alpha = 1/(T^2)$ and $\delta = 1/(2T)$ in Lemma B.5, we have the following confidence parameter that can guarantee that $f^*$ is contained in the confidence set with high probability:

$$
\beta_T(1/(2T), 1/(T^2)) = 8\sigma^2 \log\left( 2N\left(\mathcal{F}, 1/(T^2), \|\cdot\|_2\right) T \right) + 2\left( 8C + \sqrt{8\sigma^2 \log(8T^3)} \right)/T,
$$

where recall that $N\left(\mathcal{F}, \alpha, \|\cdot\|_2\right)$ is the $\alpha$-covering number of $\mathcal{F}$ with respect to the $\|\cdot\|_2$-norm. $\qquad\square$

## A.3 Proof of Theorem 5.1

*Proof.* Denote the *imagined* optimal policy $\pi_{f_t}$ under a sampled model $f_t$ as $\pi_{f_t} = \max_\pi V_\pi^{f_t}$. For PSRL, its expected regret can be decomposed as

$$
\begin{aligned}
\text{BayesRegret}(T, \pi^{\text{PSRL}}, \phi) &= \sum_{t=1}^{T} \mathbb{E}[V_{\pi^*}^{f^*} - V_{\pi_t}^{f^*}] \\
&= \sum_{t=1}^{T} \mathbb{E}[V_{\pi^*}^{f^*} - V_{\pi_{f_t}}^{f^*}] \\
&= \sum_{t=1}^{T} \mathbb{E}[V_{\pi_{f_t}}^{f_t} - V_{\pi_{f_t}}^{f^*}],
\end{aligned}
$$
(A.19)

where the second equality holds since the PSRL policy $\pi_t := \pi_{f_t}$ for a sampled $f_t$. The third equality follows from (A.9), obtained by the Posterior Sampling Lemma and the law of total expectation.

Similar with the proof in A.2, we obtain from the Simulation Lemma B.1 that

$$
\begin{aligned}
\mathbb{E}\left[\left| V_{\pi_{f_t}}^{f_t} - V_{\pi_{f_t}}^{f^*} \right|\right] &= \gamma \mathbb{E}\left[\left| \mathbb{E}_{(s,a)\sim\rho_\pi}[(f_t(\cdot|s,a) - f^*(\cdot|s,a)) \cdot V^\pi(s,a)] \right|\right] \\
&\leq \gamma \mathbb{E}\left[\left| \mathbb{E}_{(s,a)\sim\rho_\pi}[L_t \cdot \|f_t(s,a) - f^*(s,a)\|_2] \right|\right],
\end{aligned}
$$
(A.20)

where the equality follows from Lemma B.1 and the inequality follows the simulation property of continuous distributions and the Lipschitz value function assumption.

Define the event $E = \left\{ f^* \in \bigcap_t \mathcal{F}_t, f_t \in \bigcap_t \mathcal{F}_t \right\}$. The expected regret can be reduced to the sum of set widths:

$$
\begin{aligned}
\mathbb{E}\left[V_\pi^{f_t} - V_\pi^{\widetilde{f}_t}\right] &\leq \gamma \mathbb{E}\left[\left| \mathbb{E}_{(s,a)\sim\rho_\pi}\left[\mathbb{E}[L_t|E]\omega_t(s,a) + (1 - \mathbb{P}(E))C\right] \right|\right] \\
&\leq \gamma \mathbb{E}\left[\left| \mathbb{E}_{(s,a)\sim\rho_\pi}\left[L_t/(1 - 4\delta)\omega_t(s,a) + 4\delta C\right] \right|\right] \\
&\leq \gamma \mathbb{E}\left[\frac{L_t}{1 - 4\delta} \cdot \left| \mathbb{E}_{(s,a)\sim\rho_\pi}\left[\omega_t(s,a)\right] \right|\right] + 4\gamma\delta C,
\end{aligned}
$$
(A.21)

where the second inequality follows from the construction of confidence set that $\mathbb{P}\left(f^* \in \bigcap_t \mathcal{F}_t\right) \geq 1 - 2\delta$ and thus $\mathbb{P}(E) \geq 1 - 4\delta$.

Therefore, the PSRL expected regret can be bounded by

$$\text{BayesRegret}(T, \pi^{\text{PSRL}}, \phi) \leq \gamma \frac{L}{1 - 4\delta} \sum_{t=1}^{T} \mathbb{E}[\omega_t] + 4T\gamma\delta C, \tag{A.22}$$

From the proof in A.2, the expected regret of CDPO is bounded by

$$\text{BayesRegret}(T, \pi^{\text{CDPO}}, \phi) \leq \gamma \frac{L}{1 - 4\delta} \sum_{t=1}^{T} 3\mathbb{E}[\omega_t] + 8T\gamma\delta C, \tag{A.23}$$

The claim is thus established. $\qquad\square$

## B  Useful Lemmas

**Lemma B.1** (Simulation Lemma). *For any policy $\pi$ and transition $f_1$, $f_2$, we have*

$$V_\pi^{f_1} - V_\pi^{f_2} = \gamma(I - \gamma f_2^\pi)^{-1}(f_1 - f_2)V_\pi^{f_1}. \tag{B.1}$$

*Proof.* Denote the expected reward under policy $\pi$ as $r_\pi$. Let $f^\pi$ be the transition matrix on state-action pairs induced by policy $\pi$, defined as $f^\pi_{(s,a),(s',a')} := P(s'|s,a)\pi(a'|s')$.

Then we have

$$V_\pi = r_\pi + \gamma f^\pi V_\pi.$$

Since $\gamma < 1$, it is easy to verify that $I - \gamma f^\pi$ is full rank and thus invertible. Therefore, we can write

$$V_\pi = (I - \gamma f^\pi)^{-1} r_\pi. \tag{B.2}$$

Therefore, we conclude the proof by

$$\begin{aligned}
V_\pi^{f_1} - V_\pi^{f_2} &= V_\pi^{f_1} - (I - \gamma f_2^\pi)^{-1} r_\pi \\
&= (I - \gamma f_2^\pi)^{-1} \cdot \left((I - \gamma f_2^\pi) - (I - \gamma f_1^\pi)\right) V_\pi^{f_1} \\
&= \gamma(I - \gamma f_2^\pi)^{-1}(f_1^\pi - f_2^\pi) V_\pi^{f_1} \\
&= \gamma(I - \gamma f_2^\pi)^{-1}(f_1 - f_2) V_\pi^{f_1},
\end{aligned}$$

where the second equality follows from the Bellman equation. $\qquad\square$

**Lemma B.2** (Performance Difference Lemma). *For all policies $\pi$, $\pi^*$ and distribution $\mu$ over $\mathcal{S}$, we have*

$$J(\pi) - J(\pi') = \frac{1}{1 - \gamma} \cdot \mathbb{E}_{(s,a)\sim\sigma_\pi}[A^{\pi'}(s, a)]. \tag{B.3}$$

*Proof.* This lemma is widely adopted in RL. Proof can be found in various previous works, e.g. Lemma 1.16 in [3].

Let $\mathbb{P}^\pi(\tau|s_0 = s)$ denote the probability of observing trajectory $\tau$ starting at state $s_0$ and then following $\pi$. Then the value difference can be written as

$$\begin{aligned}
V_\pi^{f^*}(s) - V_{\pi'}^{f^*}(s) &= \mathbb{E}_{\tau\sim\mathbb{P}^\pi(\cdot|s_0=s)}\left[\sum_{h=0}^{\infty} \gamma^h r(s_h, a_h)\right] - V_{\pi'}^{f^*}(s) \\
&= \mathbb{E}_{\tau\sim\mathbb{P}^\pi(\cdot|s_0=s)}\left[\sum_{h=0}^{\infty} \gamma^h \left(r(s_h, a_h) + V_{\pi'}^{f^*}(s_h) - V_{\pi'}^{f^*}(s_h)\right)\right] - V_{\pi'}^{f^*}(s) \\
&= \mathbb{E}_{\tau\sim\mathbb{P}^\pi(\cdot|s_0=s)}\left[\sum_{h=0}^{\infty} \gamma^h \left(r(s_h, a_h) + \gamma V_{\pi'}^{f^*}(s_{h+1}) - V_{\pi'}^{f^*}(s_h)\right)\right]
\end{aligned}$$

Following the law of iterated expectations, we obtain

$$V_\pi^{f^*}(s) - V_{\pi'}^{f^*}(s) = \mathbb{E}_{\tau \sim \mathbb{P}^\pi(\cdot|s_0=s)} \Big[ \sum_{h=0}^\infty \gamma^h \big( r(s_h, a_h) + \gamma \mathbb{E}[V_{\pi'}^{f^*}(s_{h+1})|s_h, a_h] - V_{\pi'}^{f^*}(s_h)) \big) \Big]$$

$$= \mathbb{E}_{\tau \sim \mathbb{P}^\pi(\cdot|s_0=s)} \Big[ \sum_{h=0}^\infty \gamma^h \big( Q_{\pi'}^{f^*}(s_h, a_h) - V_{\pi'}^{f^*}(s_h)) \big) \Big]$$

$$= \mathbb{E}_{\tau \sim \mathbb{P}^\pi(\cdot|s_0=s)} \Big[ \sum_{h=0}^\infty \gamma^h A_{\pi'}^{f^*}(s_h, a_h) \Big], \tag{B.4}$$

where the third equation rearranges terms in the summation via telescoping, and the fourth equality follows from the law of total expectation.

From the definition of objective $J(\pi)$ in (2.3), we obtain

$$J(\pi) - J(\pi') = \mathbb{E}_{s_0 \sim \zeta}[V_\pi^{f^*}(s_0) - V_{\pi'}^{f^*}(s_0)]$$

$$= \frac{1}{1-\gamma} \mathbb{E}_{(s,a) \sim \sigma_\pi}[A^{\pi'}(s, a)]. \tag{B.5}$$

$\square$

**Lemma B.3** (Performance Difference and Model Error). For any two policies $\pi_1$ and $\pi_2$, it holds that

$$J(\pi_1) - J(\pi_2) = \mathbb{E}_{s \sim \zeta} \big[ V_{\pi_1}^f(s) - V_{\pi_2}^f(s) \big]$$

$$- \frac{\gamma}{2(1-\gamma)} \Big( \mathbb{E}_{\rho_{\pi_1}} \big[ \| f(\cdot|s, a) - f^*(\cdot|s, a) \|_1 \big] + \mathbb{E}_{\rho_{\pi_2}} \big[ \| f(\cdot|s, a) - f^*(\cdot|s, a) \|_1 \big] \Big).$$

*Proof.* The proof can be established by combining the Performance Difference Lemma and the Simulation Lemma. We refer to Corollary 3.1 in [48] or Lemma A.3 in [59] for a detailed proof. $\square$

**Lemma B.4** (Least Squares Generalization Bound). Given a dataset $\mathcal{H} = \{x_i, y_i\}_{i=1}^n$ where $x_i \in \mathcal{X}$ and $x_i, y_i \sim \nu$, and $y_i = f^*(x_i) + \epsilon_i$. Suppose $|y_i| \leq Y$ and $\epsilon_i$ is independently sampled noise. Given a function class $\mathcal{F} : \mathcal{X} \to [0, Y]$, we assume approximate realizable, i.e., $\min_{f \in \mathcal{F}} \mathbb{E}_{x \sim \nu} \big[ |f^*(x) - f(x)|^2 \big] \leq \epsilon_{\text{approx}}$. Denote $\widehat{f}$ as the least square solution, i.e., $\widehat{f} = \operatorname{argmin}_{f \in \mathcal{F}} \sum_{i=1}^n \big( f(x_i) - y_i \big)^2$. With probability at least $1 - \delta$, we have

$$\mathbb{E}_{x \sim \nu} \Big[ \big( \widehat{f}(x) - f^*(x) \big)^2 \Big] \leq \frac{22 Y^2 \ln(|\mathcal{F}|/\delta)}{n} + 20\epsilon_{\text{approx}}. \tag{B.6}$$

*Proof.* The result is standard and can be proved by using the Bernstein's inequality and union bound. Detailed proof can be found at Lemma A.11 in [3]. $\square$

**Lemma B.5** (Confidence sets with high probability). If the control parameter $\beta_t(\delta, \alpha)$ is set to

$$\beta_t(\delta, \alpha) = 8\sigma^2 \log(N(\mathcal{F}, \alpha, \|\cdot\|_2)/\delta) + 2\alpha t \Big( 8C + \sqrt{8\sigma^2 \log(4t^2/\delta)} \Big), \tag{B.7}$$

then for all $\delta > 0$, $\alpha > 0$ and $t \in \mathbb{N}$, the confidence set $\mathcal{F}_t = \mathcal{F}_t(\beta_t(\delta, \alpha))$ satisfies:

$$P \Big( f^* \in \bigcap_t \mathcal{F}_t \Big) \geq 1 - 2\delta. \tag{B.8}$$

*Proof.* See [43] Proposition 5 for a detailed proof. $\square$

**Lemma B.6** (Bound of Set Width Sum). If $\{\beta_t | t \in \mathbb{N}\}$ is nondecreasing with $\mathcal{F}_t = \mathcal{F}_t(\beta_t)$ and $\|f\|_2 \leq C$ for all $f \in \mathcal{F}$, then finite-horizon MDP we have

$$\sum_{t=1}^T \sum_{h=1}^H \omega_t(s_h, a_h) \leq 1 + HC \dim_E(\mathcal{F}, T^{-1}) + 4\sqrt{\dim_E(\mathcal{F}, T^{-1})\beta_T T}, \tag{B.9}$$

where $\omega_t(s, a) = \sup_{\underline{f}, \overline{f} \sim \mathcal{F}_t} \|\overline{f}(s, a) - \underline{f}(s, a)\|_2$.

*Proof.* See [43] Proposition 6 for a detailed proof. $\square$

## C  Limitations of Eluder Dimension

In Theorem 5.8, the *eluder dimension* $d_E$ appears in the Bayes expected regret bound to capture how effectively the observed samples can extrapolate to unobserved transitions.

For some specific function classes, Osband et al. [43] provide the corresponding eluder dimension bound, e.g., for (generalized) linear function classes, quadratic function class, and for finite MDPs, c.f. Proposition 1-4 in [43].

However, for non-linear models, Dong et al. [13] show that the $\varepsilon$-eluder dimension of one-layer neural networks is *at least* exponential in model dimension. Similar results are also established in [33]. We refer to Section 5 in [13] or Section 4 in [33] for details and more explanations.

## D  Additional Related Work

Some MBRL work also concerns iterative policy improvement. SLBO [35] provides a trust-region policy optimization framework based on OFU. However, the conditions for monotonic improvement cannot be satisfied by most parameterized models [35, 13], which leads to a greedy algorithm in practice. Prior work that shares similarities with ours contains DPI [59] and GPS [31, 39] as dual policy optimization procedures are adopted. Both DPI and GPS leverage a *locally* accurate model and use different objectives for imitating the intermediate policy within a trust-region. However, the policy imitation procedure updates the policy parameter in a *supervised* manner, which poses additional challenges for effective exploration, resulting in unknown convergence results even with a simple model class. In contrast, CDPO by taking the epistemic uncertainty into consideration can be shown to achieve global optimality. In fact, greedy model exploitation is provably optimal only in very limited cases, e.g., linear-quadratic regulator (LQR) settings [36].

OFU-RL has shown to achieve an optimal sublinear regret when applied to online LQR [1], tabular MDPs [19] and linear MDPs [22]. Among them, HUCRL [10] is a deep algorithm proposed to deal with the joint optimization intractability in (3.1). Besides, Russo and Van Roy [49, 50] unify the bounds in various settings (e.g., finite or linear MDPs) by introducing an additional model complexity measure — eluder dimension. Other complexity measure include witness rank [60], linear dimensionality [66] and sequential Rademacher complexity [13].

## E  Algorithm Instantiations

The model-based policy optimization solver $\texttt{MBPO}(\pi, \{f\}, \mathcal{J})$ in Algorithm 1 can be instantiated as one of the following algorithms, Dyna-style policy optimization in Algorithm 2, model-based back-propagation in Algorithm 3, and model predictive control policy optimization in Algorithm 4. By default, $\texttt{MBPO}$ is instantiated as the Dyna solver (i.e. Algorithm 2) in our MuJoCo experiments and as the policy iteration solver in our $N$-Chain MDPs experiments. We note that the instantiations are not restricted to the listed algorithms, and many other $\texttt{MBPO}$ algorithms that augment policy learning with a predictive model can also be leveraged, e.g., model-based value expansion [15, 6]. In the Referential Update step where no input policy exists in $\texttt{MBPO}(\cdot, \widehat{f}_t^{LS}, (4.1))$, we initialize policy $\pi = \pi_{t-1}$, i.e. the reactive policy from the last iteration.

**Dyna.** Dyna involves model-generated data and optimizing the policy with any model-free RL method, e.g., REINFORCE or actor-critic [28]. The state-action value can be estimated by learning a critic function or unrolling the model. In Constrained Conservative Update, the input objective function $\mathcal{J}$ is (4.2), which is with constraints. Thus, the Lagrangian multiplier is introduced, similar to the model-free trust-region algorithms [53, 54, 2].

**Back-Propagation Through Time.** BPTT [30, 64] is a first-order model-based policy optimization framework based on pathwise gradient (or reparameterization gradient) [58]. There are also several variants including Stochastic Value Gradients (SVG) [18], Model-Augmented Actor-Critic (MAAC) [9], and Probabilistic Inference for Learning COntrol (PILCO) [12]. Specifically, the policy parameters are updated by directly computing the derivatives of the performance with respect to the parameters. When the optimization of objective function is constrained, the accumulating step (Algorithm 3

---

**Algorithm 2** Dyna Model-Based Policy Optimization

---

**Input:** Policy $\pi$, model set $\{f\}$, objective function $\mathcal{J}$.

1: Initialize a simulation data buffer $\widehat{\mathcal{D}}$
2: Sample a batch of initial states from the initial distribution $\zeta$
3: ▷ Data simulation
4: **for** initial state sample $s_0$ **do**
5:     **for** model $f$ in model set $\{f\}$ **do**
6:         **for** timestep $h = 1, ..., H$ **do**
7:             Sample action $\widehat{a}_h \sim \pi(\cdot|\widehat{s}_h)$
8:             Sample simulation state $\widehat{s}_{h+1} \sim f(\widehat{s}_h, \widehat{a}_h)$
9:             Append simulation data to buffer $\widehat{\mathcal{D}} = \widehat{\mathcal{D}} \cup (\widehat{s}_h, \widehat{a}_h, r_h, \widehat{s}_{h+1})$
10:         **end for**
11:     **end for**
12: **end for**
13: ▷ Policy optimization with any model-free algorithm `ModelFree`
14: Objective optimization of policy on the simulated data $\pi \leftarrow \texttt{ModelFree}(\widehat{\mathcal{D}}, \pi)$

---

Line 9) can be $L \leftarrow L + \gamma^h r(\widehat{s}_h, \widehat{a}_h) - \lambda D_{\text{KL}}$, where $\lambda$ is the Lagrangian multiplier and $D_{\text{KL}}$ is the corresponding KL constraint.

---

**Algorithm 3** Model-Based Back-Propagation Policy Optimization

---

**Input:** Policy $\pi$, model set $\{f\}$, objective function $\mathcal{J}$.

1: Initialize a simulation data buffer $\widehat{\mathcal{D}}$
2: Start from initial state $s_0$
3: Reset $L \leftarrow 0$
4: ▷ Data simulation
5: **for** model $f$ in model set $\{f\}$ **do**
6:     **for** timestep $h = 1, ..., H$ **do**
7:         Sample action $\widehat{a}_h \sim \pi(\cdot|\widehat{s}_h)$
8:         Sample simulation state $\widehat{s}_{h+1} \sim f(\widehat{s}_h, \widehat{a}_h)$
9:         Accumulate reward and constraint to $L$
10:     **end for**
11: **end for**
12: ▷ Policy optimization
13: Compute policy gradient with back-propagation through time
14: Objective optimization of policy $\pi \leftarrow \texttt{PolicyGradient}$

---

**Model Predictive Control Policy Optimization.** MPC is a *planning* framework that directly generates optimal action sequences under the model. Different from the above model-augmented policy optimization methods, MPC policy optimization directly generates optimal action sequences under the model and then distills the policy. Specifically, the pseudocode in Algorithm 4 begins with initial actions generated by the policy. Then with a shooting method, e.g., the cross-entropy method (CEM), the actions are refined and the policy that generates these optimal actions are distilled. Below, the algorithm to obtain the refined actions `EliteActions` can be CEM with action noise added to the action or policy parameter, i.e., POPLIN-A and POPLIN-P in [63]. The policy can be updated by `UpdatePolicy` using behavior cloning.

**Policy Iteration for Tabular MDPs.** In tabular settings where the state space $\mathcal{S}$ and action space $\mathcal{A}$ are discrete and countable, we can perform policy iteration under each model in the model set $\{f\}$. Here, the model is the tabular representation instead of function approximators. Based on the state-action values under various models, the optimal action at each state is the one that maximizes the weighted average of the values within the constraint of total variation distance.

**Algorithm 4** Model Predictive Control Policy Optimization

**Input:** Policy $\pi$, model set $\{f\}$, objective function $\mathcal{J}$, algorithm to update actions `EliteActions`, algorithm to update policy `UpdatePolicy`.

1: Start from initial state $s_0$
2: Reset $J \leftarrow 0$
3: ▷ Model-based planning
4: **for** model $f$ in model set $\{f\}$ **do**
5:    **for** timestep $h = 1, ..., H$ **do**
6:       Sample action $\widehat{a}_h \sim \pi(\cdot|\widehat{s}_h)$
7:       Sample simulation state $\widehat{s}_{h+1} \sim f(\widehat{s}_h, \widehat{a}_h)$
8:       Accumulate reward and constraint to $J$
9:    **end for**
10: **end for**
11: $\mathbf{a} \leftarrow$ `EliteActions`$(J, \widehat{a}_{1:N})$
12: ▷ Policy distillation
13: $\pi \leftarrow$ `UpdatePolicy`$(\mathbf{a})$

## F    Experimental Settings and Results in $N$-Chain MDPs

### F.1    Settings of MuJoCo Experiments

In the MuJoCo experiments, we use a 5-layer neural network to approximate the dynamical model. We use deterministic ensembles [8] to capture the model epistemic uncertainty. Specifically, different ensembles are learned with independent transition data to construct the 1-step ahead confidence interval at every timestep. Each ensemble is separately trained using Adam [26]. And the number of ensemble heads can be set to 3, 4, or, 5, each of which is shown to be able to provide considerable performance in our experiments. All the experiments are repeated with 6 random seeds.

Since neural networks are not calibrated in general, i.e., the model uncertainty set is not guaranteed to contain the real dynamics, we follow HUCRL [10] to re-calibrate [29] the model. Our MuJoCo code is also built upon the HUCRL GitHub repository.

When using the Dyna model-based policy optimization, the number of gradient steps for each optimization procedure in an iteration is set to 20. And we empirically find that the KL divergence (or total variance) constraint makes the algorithm more efficient when computing the argmax in the optimization step, since optimizing from $\pi_{t-1}$ at iteration $t$ needs fewer policy gradient steps if the policy update is constrained within a certain trust region.

The task-specific and task-common settings and parameters are listed below in Table 1.

Table 1: Experimental parameters.

|  | Inverted Pendulum | Pusher | Half-Cheetah |
|---|---|---|---|
| episode length $H$ | 200 | 150 | 1000 |
| dimension of state | 4 | 23 | 18 |
| dimension of action | 1 | 7 | 6 |
| action penalty | 0.001 | 0.1 | 0.1 |
| hidden nodes | (200, 200, 200, 200, 200) | | |
| activation function | Swish | | |
| optimizer | Adam | | |
| learning rate | $10^{-3}$ | | |

### F.2    Experiments in $N$-Chain MDPs

Besides the experiments in MuJoCo, we also conduct tabular experiments in the $N$-Chain environment that is proposed in [37]. Specifically, there are in total 2 actions and $N$ states in an MDP. The initial state is $s_1$ and the agent can choose to go *left* or *right* at each of the $N$ states. The *left* action always succeeds and moves the agent to the left state, giving reward $r \sim \mathcal{N}(0, \delta^2)$. Taking the *right* action at

state $s_1, \ldots, s_{N-1}$ gives reward $r \sim \mathcal{N}(-\delta, \delta^2)$ and succeeds with probability $1 - 1/N$, moving the agent to the right state and otherwise moving the agent to the left state. Taking the *right* action at $s_N$ gives reward $r \sim \mathcal{N}(1, \delta^2)$ and moves the agent back to $s_1$ with probability $1 - 1/N$.

We set $\delta = 0.1 \exp(-N/4)$, such that going *right* is the optimal action at least up to $N = 40$. As the number of states $N$ is increasing, the agent needs *deep* exploration (e.g. guided by uncertainty) instead of *dithering* exploration (e.g. epsilon-greedy exploration), such that the agent can keep exploring despite receiving negative rewards [45].

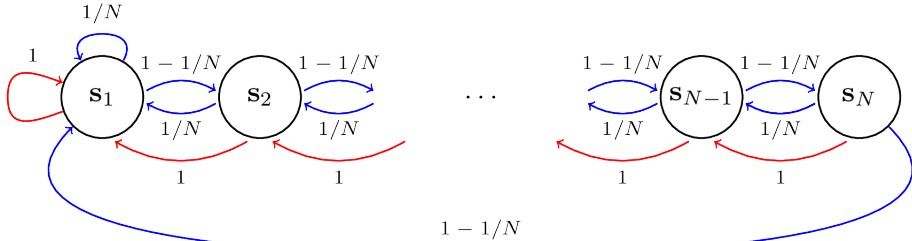

Figure 6: Illustration of the $N$-Chain MDP. Blue arrows correspond to action *right* (optimal) and red arrows correspond to action *left* (suboptimal). The figure is copied from [37].

For this reason, we evaluate the proposed algorithm CDPO and compare it with other Bayesian RL algorithms, including Bayesian Q-Learning (BQL) [11], Posterior Sampling for RL (PSRL) [42], the Uncertainty Bellman Equation (UBE) [46] and Moment Matching (MM) approach [37]. For CDPO, the dual optimization steps are solved by policy iteration, and the conservative update is performed within the total variation distance $\eta = 0.2$ (c.f. Policy Iteration for Tabular MDPs in Appendix E). We choose conjugate priors to represent the posterior distribution: we use a Categorical-Dirichlet model for discrete transition distribution at each $(s, a)$, and a Normal-Gamma (NG) model for continuous reward distribution at each $(s, a, s')$.

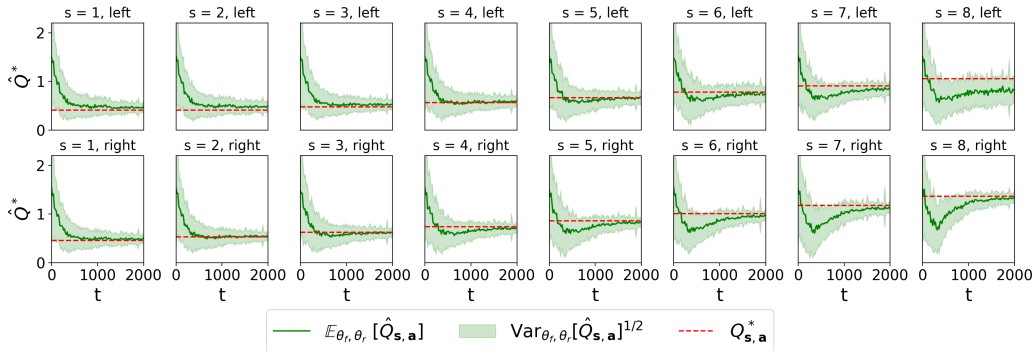

Figure 7: Posterior evolution of CDPO algorithm in the 8-Chain MDP.

**Evolution of Posterior.** Figure 7 demonstrates the evolution of the posterior of the CDPO algorithm in an 8-Chain MDP. As training progresses the posteriors concentrate on the true optimal state-action values and the behavior policy converges on the optimal one. The fast reduction of uncertainty is central to achieving principled and efficient exploration.

Compared to the posterior evolution of the PSRL algorithm corresponding to the optimal actions, i.e. the bottom row of curves in Figure 8, the expected value estimates of CDPO are closer to the ground-truth, and the variance is also smaller. Notably, the variance of CDPO might be higher for suboptimal actions, e.g., $s = 8, a = \text{left}$ (the last image of the first row in Figure 7). It is due to the conservative nature of CDPO that it only cares about the *expected* value, instead of the value of a sampled (imperfect) model as in PSRL. In other words, as long as the uncertainty is large, the PSRL agents can take suboptimal actions to explore the uninformative regions, which causes the inefficient over-exploration issue.

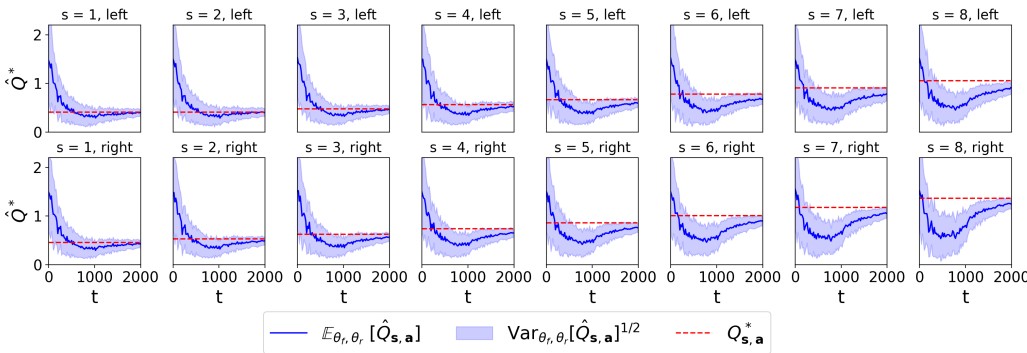

Figure 8: Posterior evolution of PSRL algorithm in the 8-Chain MDP.

**Cumulative Regret.** We compare CDPO and previous algorithms on the $N$-Chain MDPs with various state sizes $N$ by measuring the cumulative regret of an oracle agent following the optimal policy. The results are shown in Figure 9. To make the performances comparable on the same scale, we also provide the normalized regret in Figure 10.

We observe that when the size of state space $N$ is relatively smaller, e.g. $N \leq 5$, CDPO, PSRL, BQL, and MM algorithms achieve sublinear regret. The performances of these algorithms are also comparable, showing the necessity of deep exploration. On the contrary, Q-Learning which only relies on dithering exploration mechanisms fail to find the optimal strategy. However, as $N$ is increasing, where the exploration must be effective for the agent to continually explore despite receiving negative rewards, the CDPO agents offer significantly lower cumulative regret and faster convergence.

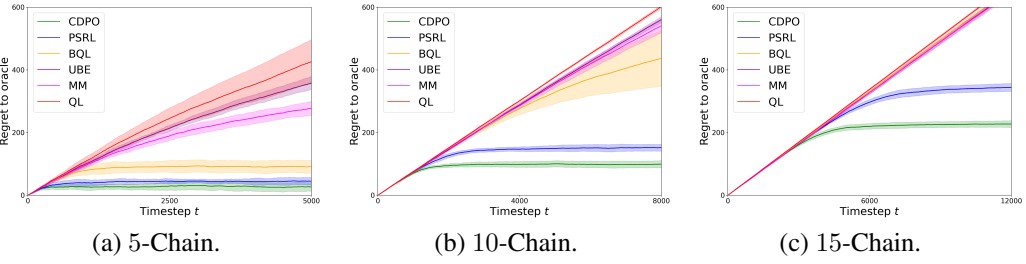

| (a) 5-Chain. | (b) 10-Chain. | (c) 15-Chain. |

Figure 9: Comparison of cumulative regret.

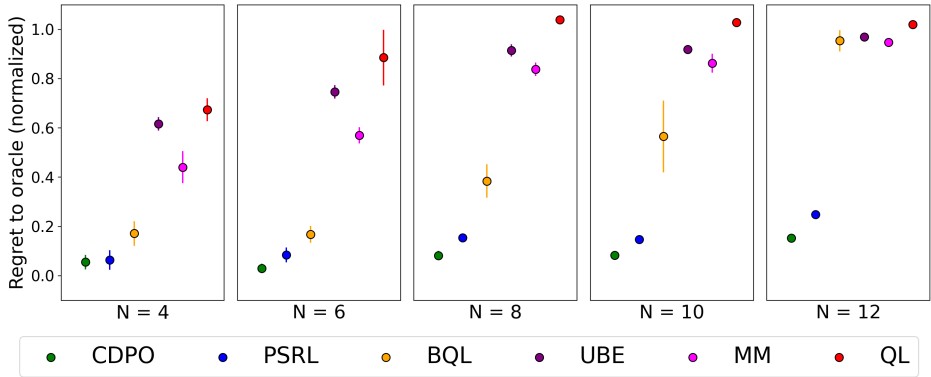

Figure 10: Performance comparison in terms of regret to the oracle.

# G  Algorithmic Comparisons between MBRL Algorithms

We provide algorithmic comparisons of four MBRL frameworks, including greedy model exploitation algorithms, OFU-RL, PSRL, and the proposed CDPO algorithm.

The differences mainly lie in the model selection and policy update procedures. The high-level pseudocode is given in Algorithm 5, 6, 7 and 8. Among them, the greedy model exploitation algorithm is a naive instantiation, where other instantiations can include the ones that augment Algorithm 5 with e.g., a dual framework that involves a locally accurate model and a supervised imitating procedure [59, 31]. In Algorithm 5, $\widetilde{f}_t$ can either be a probabilistic model or a deterministic model (with additive noise), which can be estimated via Maximum Likelihood Estimation (MLE) or minimizing the Mean Squared Error (MSE), respectively.

---

**Algorithm 5** Naive Greedy Model Exploitation

1: **for** iteration $t = 1, ..., T$ **do**
2:    Estimate model $\widetilde{f}_t$ via MLE or MSE
3:    Compute $\pi_t = \mathrm{argmax}_\pi V_\pi^{\widetilde{f}_t}$
4:    Execute $\pi_t$ in the real MDP
5:    $\mathcal{H}_{t+1} = \mathcal{H}_t \cup \{s_{h,t}, a_{h,t}, s_{h+1,t}\}_h$
6: **end for**
7: **return** policy $\pi_T$

---

**Algorithm 6** OFU-RL Algorithm

1: **for** iteration $t = 1, ..., T$ **do**
2:    Construct confidence set $\mathcal{F}_t$
3:    Compute $\pi_t = \mathrm{argmax}_{\pi, f \sim \mathcal{F}_t} V_\pi^{f_t}$
4:    Execute $\pi_t$ in the real MDP
5:    $\mathcal{H}_{t+1} = \mathcal{H}_t \cup \{s_{h,t}, a_{h,t}, s_{h+1,t}\}_h$
6: **end for**
7: **return** policy $\pi_T$

---

**Algorithm 7** PSRL Algorithm

1: **for** iteration $t = 1, ..., T$ **do**
2:    Sample $f_t \sim \phi(\cdot \mid \mathcal{H}_t)$
3:    Compute $\pi_t = \mathrm{argmax}_\pi V_\pi^{f_t}$
4:    Execute $\pi_t$ in the real MDP
5:    $\mathcal{H}_{t+1} = \mathcal{H}_t \cup \{s_{h,t}, a_{h,t}, s_{h+1,t}\}_h$
6: **end for**
7: **return** policy $\pi_T$

---

**Algorithm 8** CDPO Algorithm

1: **for** iteration $t = 1, ..., T$ **do**
2:    Referential Update $q_t$ following (4.1)
3:    Conservative Update $\pi_t$ following (4.2)
4:    Execute $\pi_t$ in the real MDP
5:    $\mathcal{H}_{t+1} = \mathcal{H}_t \cup \{s_{h,t}, a_{h,t}, s_{h+1,t}\}_h$
6: **end for**
7: **return** policy $\pi_T$

---

# H  Societal Impact

For real-world applications, interactions with the system imply energy or economic costs. With practical efficiency, CDPO reduces the training investment and is aligned with the principle of responsible AI. However, as an RL algorithm, CDPO is unavoidable to introduce safety concerns, e.g., self-driving cars make mistakes during RL training. Although CDPO does not explicitly address them, it may be used in conjunction with safety controllers to minimize negative impacts, while drawing on its powerful MBRL roots to enable efficient learning.