# OpenReview forum: "Conservative Dual Policy Optimization for Efficient Model-Based Reinforcement Learning"
_NeurIPS.cc/2022/Conference — NeurIPS 2022 Accept_

### Official Review · Reviewer_q3JM · 2022-07-09

**Rating:** 5
**Confidence:** 4
**Soundness:** 2 fair
**Presentation:** 2 fair
**Contribution:** 2 fair

**Summary:**

The paper proposes Conservative Dual Policy Optimization (CDPO) that involves a Referential Update and a Conservative Update. It is proven that CDPO achieves the same regret as PSRL and ensures the monotone improvement of the policy. Experimental results validates its effectiveness.



**Questions:**

See the weakness.

**Ethics Review Area:**

["I don’t know"]

**Strengths And Weaknesses:**

Strengths:
1.The proposed method is simple and effective. Theoretical results are sufficient.
2. The literature review is sufficient.

Weakness:
1. The motivation of the proposed method is not clear and the paper is not easy to follow.
2. The paper only test their algorithm on 3 environments of MuJoCo. I am curious about the performance on larger env such as Ant and Humanoid.
3. I am also interested at the improvement ratio of sample efficiency over baselines.

---

> ### Author Response · Authors · 2022-08-02
> **Initial response to Reviewer q3JM**
>
> We thank the reviewer for identifying our technical contributions. We have improved our submission (marked blue in the revision) based on the review’s valuable comments. Below are our specific responses to the concerns raised by the reviewer:
>
> ---
>
> #### **1. Our motivation.**
> In our revision, we make two changes in **Section 3** and **Section 6** to help better understand our motivation:
> - First, we move **Theorem 3.1** from the Appendix to the main body, which states that the eluder dimension of nonlinear models cannot be polynomially bounded, a result established in previous work. The theorem indicates the limitation of previous provably efficient algorithms with regret $O(\sqrt{d_E T})$: additional complexity is hidden in the eluder dimension, e.g. when $\varepsilon = T^{-1}$, the regret contains $d_E = \Omega(T^{d-1})$ and is no longer sublinear in $T$, thus losing the desired property of global optimality and sample efficiency. This is the underlying reason for the over-exploration issue that we aim to address with a dual framework containing two basic designs: choosing a _stable_ reference model and maximizing the _expected_ value.
> - We also move part of the tabular experimental results to **Section 6.1** for a better understanding of the difference between previous provable algorithms and the proposed CDPO. We highlight our observation here: CDPO gives more accurate and certain estimations _only_ for the optimal actions (still has better regret), while PSRL puts efforts into trying suboptimal actions. This verifies the potential over-exploration issue in PSRL: as long as the uncertainty contains unrealistically large values, PSRL agents can perform uninformative exploration by acting suboptimally according to an inaccurate sampled model. In contrast, CDPO replaces the sampled model with a _stable_ mean estimate and cares about the _expected_ value, thus avoiding such pitfalls.
>
> ---
>
> #### **2. Performance in larger environments.**
> - As suggested by the reviewer, we have included additional experiment results in the Ant task in our revision (**Fig. 3(d)** and **4(d)**).
>
> ---
>
> #### **3. Improvement ratio of sample efficiency over baselines.**
> We add comparisons of the sample efficiency of the proposed algorithm and the model-based RL baselines in **Appendix F.2**. Specifically, we report the required number of samples to reach $6000$ in the half-cheetah task. Then we list the sample efficiency ratio and runtime ratio, i.e., how many times samples and running time the baselines need to reach the same performance as CDPO. We highlight our results on the sample efficiency here:
>
> |          | Sample Number | Sample Efficiency Ratio |
> |----------|---------------|-------------------------|
> | CDPO     | 276K          | 1                       |
> | PSRL     | 908K          | 3.3                     |
> | HUCRL    | 1137K         | 4.1                     |
> | DPI      | 880K          | 3.2                     |
> | MBPO     | 1014K         | 3.7                     |
>
> We can observe from the table that CDPO needs fewer samples to reach the same result compared to other baselines, thus having higher sample efficiency. See Appendix F.2 for the full results.
>
> ---
>
> We hope the reviewer can consider raising the score if we resolved the reviewer's concerns. We would be happy to have further discussions if the reviewer has any additional questions or comments.

---

> > ### Comment · Reviewer_q3JM · 2022-08-10
> > **Thanks for the response**
> >
> > I am pleased with your detailed response.

---

### Official Review · Reviewer_M8qo · 2022-07-10

**Rating:** 7
**Confidence:** 3
**Soundness:** 3 good
**Presentation:** 3 good
**Contribution:** 3 good

**Summary:**

This paper proposes a novel algorithm: Conservative Dual Policy Optimization (CDPO). CDPO involves a referential update optimized optimized for a reference model, which is similar to PSRL. It also involves a conservative update, which provides a conservative range of randomness by maximizing the expectation of model value. CDPO provides same regret bound as of PSRL and can simultaneously offer monotonic policy improvement and global optimality. The paper presents this CDPO framework along with theoretical and empirical analysis.

**Questions:**

The paper is well written and well organized, answering most of the questions up front. Thanks to the authors.

Section 2 of the paper builds up the technical ground for model based RL and presents the basic for regret analysis. Section 3 explains OFU and PSRL framework along with general complexity and performance bound analysis for them.

The algorithm presented in section 4 is concise and clear. Theoretical analysis presented in section 5 is solid.

Empirical evaluation is performed on multiple domains and compared against different SOTA models. The CDPO algorithm shows clear evidence of performing better, specially for domain with higher dimensional state space involving more uncertainties.

**Ethics Review Area:**

["I don’t know"]

**Limitations:**

Yes

**Strengths And Weaknesses:**

The paper presents a novel algorithm that can better handle exploration and learning in the face of uncertainty. The contribution is solid given the simplicity, theoretical soundness and empirical superiority of the proposed method.

---

> ### Author Response · Authors · 2022-08-02
> **Initial response to Reviewer M8qo**
>
> We appreciate that our paper is recognized for several positive aspects:
> - Solid contribution with comprehensive analysis.
> - A simple yet effective framework.
> - Theoretical soundness and empirical superiority.
> - Clear writing and good motivation.
>
> We sincerely thank the reviewer for the meticulous reading and inspirational comments.

---

### Official Review · Reviewer_2poR · 2022-07-11

**Rating:** 7
**Confidence:** 4
**Soundness:** 3 good
**Presentation:** 4 excellent
**Contribution:** 4 excellent

**Summary:**

This work addresses the problem of model-based planning in RL, focused on nonlinear models and high-dimensional environments. As the model in these cases tends to be erroneous, planning greedily or optimistically wrt the model can lead to poor performance. The authors propose a conservative update algorithm that relies on greedy updates of a reference policy wrt the least-squares estimate of the model, and another update of the behaviour policy that is constrained to remain close to the reference policy but also to optimize the expected performance wrt. the posterior over models. Theoretically, they show that greedy optimization wrt the least-squares model mimics the behaviour of PSRL in terms of Expected Regret. They also bound the policy improvement at each iteration, and provide global convergence results for classes of models with finite complexity measures. Empirical evaluations show that their proposed algorithm (CDPO) has improved performance over some common baselines on a few continuous control domains.


**Questions:**

- Each optimization in (4.1) and (4.2) needs to be done to completion. In the deep learning regime, these optimizations may never converge to the optimum point or even good local optima. How does this impact the results? Has this been an issue in the experiments and how was it addressed?

Section 6.1:
- L309-310: “incompatibility brought by the model”: Does this refer to the model class not being sufficient to learn a good model of the environment? If so, how do we know this network is not able to learn a good model of the environment for HalfCheetah? It would help to see the model prediction loss for example. Otherwise, I don’t think we can necessarily make this conclusion. It could be that the policy is harder to learn in HalfCheetah.
- Related to the above point — L310-311 : “This phenomenon is most obvious in the half-cheetah task, where learning a globally accurate model is hard.” and L312-313: “When the model uncertainty is large, the aggressive policy updates and uninformative exploration lead to slow convergence and lower asymptotic value within finite iterations.” This is more of a hypothesis that’s not really being shown in the results. I don’t think it can be concluded from the present results that this is why the performance is lower.
- How do we know these results are due to better exploration and not, for example, better optimization?
- Potentially a better argument for better exploration may be better shown in a finite domain where the exploration of the algorithms (i.e. what states are visited) can be visualized.


**Limitations:**

I did not find that the authors sufficiently discussed the limitations of their work. See Questions.

**Strengths And Weaknesses:**

This paper is very well-written and easy to follow. The authors do a good job of placing it within the context of existing literature.

Strengths:
- I think the idea behind CDPO is a good one, and addresses an important problem. I also feel that overall the paper does a good job of analyzing this algorithm from different perspectives (including the results in the appendix)

Weaknesses:
- Experimental results are good but could include more domains.
- Results depend on convergence of (4.1) and (4.2) (see Questions)
- I think more in-depth description and implications of the result in (5.6) would have been helpful

Regarding writing:
- L217 should be argmax
- L218-219: instead of “PSRL proceeds by …” I would suggest writing out the reasoning and the lemma that is used to obtain this result (e.g. Lemma 1 of [1]).
- L299-300: I would say “discussed” rather than “analyzed”

[1] Osband, Ian, Daniel Russo, and Benjamin Van Roy. "(More) efficient reinforcement learning via posterior sampling." Advances in Neural Information Processing Systems 26 (2013).

---

> ### Author Response · Authors · 2022-08-02
> **Initial response to Reviewer 2poR (Part 1/2)**
>
> We thank the reviewer for identifying our technical contributions and the significance of our work. The valuable comments have helped us improve our submission (marked blue in the revision). Below are our specific responses to the questions raised by the reviewer:
>
> ---
>
> #### **1. Experimental results are good but could include more domains.**
> In the updated revision, we make the following changes to the experiments.
> - We move some of our experimental results in the $N$-Chain MDP from the Appendix to the main paper **Section 6.1**, in order to highlight the insights behind the CDPO exploration and to provide evidence of why CDPO can be more efficient compared to other exploration mechanisms such as PSRL.
> - We add the evaluation of uncertainty elimination by plotting the uncertainty scale of CDPO compared to other provable methods in the half-cheetah task. The results are shown in **Appendix F.3**.
> - Besides, we conduct experiments in the higher-dimensional MuJoCo ant task. The results are shown in **Figure 3(d)** and **4(d)**.
>
> ---
>
> #### **2. The argmax operator in Eq. (4.1), (4.2) and its impact.**
> We thank the reviewer for pointing this out. The discussion below is added to the revision (marked blue on **page 5**).
>
> - The reviewer is correct that each update procedure contains an inner loop of optimization steps. Therefore, we follow previous work and assume access to an optimization oracle, which returns an optimal policy under a given model. It is worth noting that most iterative RL algorithms need such an assumption for obtaining theoretical guarantees, e.g., DPI [1], SLBO[2], TRPO [3] for monotonic improvement, and PSRL [4], HUCRL [5] for global optimality.
> - **Impact from the theory side:** In practice, the problem of finding an optimal policy under a model can be approximately solved by model-based solvers (i.e. the MBPO solver in our pseudocode), e.g. Dyna and MPC (see Sec. 4.2 and Appendix E for more details). The off-the-shelf theorems of policy gradient and MPC can be applied to obtain a more fine-grained analysis for specific policy or model classes, which, however, is beyond the scope of this paper.
> - **Impact from the experiment side:** Our ablation study in Sec. 6.4 shows that no clear gap exists between different solvers. This indicates that within each iteration, several optimization steps based on the last iteration (commonly adopted in previous work) suffice to give good results, bypassing the stringent requirement to run to convergence and reach the optima in every step. Partial reasons could be that we are not training from scratch in each iteration and the model is consistent between successive iterations.
>
>
> ---
>
> #### **3. Description and implications of the result in (5.6).**
> - When the regret is sublinear in $T$, we conclude that the actual policy converges to the optimal policy, with rates related to the order of $T$. Together with the per-iteration sample number, the sample complexity is also directly implied, since we can use regret to calculate the required iterations to reach a certain suboptimality threshold. The Bayesian regret is simply the regret placed inside of an expectation conditioned over model instances and is thus also of theoretical interest. As we care about sublinear regret to conclude global optimality (i.e. the policy value is global optimal after sufficient iterations), we go one step further from (5.6) to Corollary 5.7 to get a clear dependence of regret on $T$.

---

> > ### Author Response · Authors · 2022-08-02
> > **Initial response to Reviewer 2poR (Part 2/2)**
> >
> > #### **4. Clarification on experiments.**
> > - **“incompatibility brought by the model”:** It refers to the fact that the more-commonly-used nonlinear models do not have polynomially bounded eluder dimension (thus not theoretically efficient), while linear models (which are theoretically efficient) have poor performance in the high-dimensional tasks. We thank the reviewer for raising the confusion and we have made changes in the revision.
> > - **Hypothesis of model uncertainty:** We have improved writing and made two major changes in experiments following the review's suggestion:
> >     - To better understand the difference between the proposed CDPO and previous provable exploration mechanisms as well as to experimentally support our analysis, we move some of our existing results in $N$-Chain MDPs from the Appendix to the main paper **Section 6.1**. We highlight our observation here: CDPO gives more accurate and certain estimations _only_ for the optimal actions (still has better regret), while PSRL puts efforts into trying suboptimal actions. This verifies the potential over-exploration issue in PSRL: as long as the uncertainty contains unrealistically large values, PSRL agents can perform uninformative exploration by acting suboptimally according to an inaccurate sampled model. In contrast, CDPO replaces the sampled model with a _stable_ mean estimate and cares about the _expected_ value, thus avoiding such pitfalls.
> >     - We also add experiments on the elimination of uncertainty scale in **Appendix F.3**. We observe that the uncertainty in half-cheetah is indeed slower eliminated by PSRL and HUCRL agents.
> >
> > ---
> >
> > We have also corrected typos and improved writing, thanks to the review’s constructive suggestion. We hope the above response resolves your questions and we would be happy to have further discussions if you have any additional questions or comments.
> >
> > ---
> >
> > [1] Sun et al. "DPI." NeurIPS 2018.\
> > [2] Luo et al. "SLBO." ICLR 2019.\
> > [3] Schulman et al. "TRPO." ICML 2015.\
> > [4] Osband et al. "(More) Efficient RL via PS." NeurIPS 2013.\
> > [5] Curi et al. "HUCRL." NeurIPS 2020.

---

> > > ### Comment · Reviewer_2poR · 2022-08-05
> > > **Revisions and response answered my questions**
> > >
> > > Thank you for the insightful and thorough response. The additional explanations and experiments (particularly Figure 1) in the revision are very helpful in answering my questions.
> > > One last question was that I'm not sure how to interpret Figure 6 in Appendix F.3, but this is a low-priority question.

---

> > > > ### Author Response · Authors · 2022-08-06
> > > > **Author Response to Reviewer 2poR**
> > > >
> > > > We thank the reviewer for the reply. We appreciate that the reviewer acknowledged our additional explanations and experiments.
> > > >
> > > > - We add Appendix F.3 to better understand the exploration efficiency of different mechanisms in the higher-dimensional tasks, which complements the results of tabular settings in Section 6.1. Although the tabular results demonstrate the motivation of CDPO and the limitation of PSRL, algorithms in finite MDPs by trying every possible action can finally obtain an accurate _high-confidence_ prediction. However, it is not the case for tasks beyond finite settings, where an exploration step can in principle only eliminate an exponentially small portion of the uncertainty. So we provide evidence of the lower efficiency of previous provable exploration mechanisms in high-dimensional tasks: In Figure 6, the average uncertainty scale on the validation data of previous algorithms decreases slower. In other words, these agents are taking (suboptimal) actions that are less informative, i.e. less helpful for high-confidence estimations and the quality of future actions. (Intuitively, if the actions are suboptimal, we expect these actions can at least provide confidence of which regions are not worth exploring, so that agents can take good actions sooner.)
> > > >
> > > > We hope the above response resolves your questions.

---

### Official Review · Reviewer_NcmH · 2022-07-12

**Rating:** 5
**Confidence:** 5
**Soundness:** 3 good
**Presentation:** 3 good
**Contribution:** 3 good

**Summary:**

This paper aims to solve the over-exploration issue of posterior sampling (PSRL) in model-based reinforcement learning. They propose Conservative Dual Policy Optimization (CDPO) which involves a Referential Update and a Conservative Update. They also proves CDPO can achieves the same regret as PSRL, and experimental results show CDPO can achieve better performance than PSRL.

**Questions:**

See  Weaknesses above.

**Limitations:**

Lack of motivation, limited evaluation

**Strengths And Weaknesses:**

Strengths:
1. The paper is well-written and easy to follow.
2. They provide a detailed theoretical analysis for CDPO.
3. CDPO outperforms PSRL on three MuJoCo environments.

Weaknesses:
1. In line 7 the paper claims that ``The sampled model that current policy is greedily optimized upon will thus be unsettled, resulting in aggressive policy updates and over-exploration”. However, in the main sections, the paper doesn’t provide any theoretical analysis or empirical evidence to support this claim. They only cite two papers in line 31. This leads to the lack of reasonable motivation for the proposed method. I hope the authors can provide more detailed analysis for the over-exploration issue.
2. In Eq.4.2, CDPO uses KL divergence $\mu$ to trade off exploitation and exploration, so this is a very important hyperparameter. In section 6.3, the results show that a time-decay $\mu$ performs worse than a fixed $\mu$. This is quite strange because intuitively, as the environment is gradually explored, we should make the current policy closer to the exploitation policy. I think the ablation study for $\mu$ should be more complete and have a more specific explanation. For example, how should the time decay $\mu$ be designed, and how much does $\mu$ affect the performance of the policy.
3. Since CDPO solves two model-based solvers (line 2 and line 4 in Algorithm1), the time complexity of the algorithm should be twice that of PSRL. Many MBRL algorithms take a very long time to run to convergence (for example, one of the state-of-the-art methods MBPO [1] needs almost 48 hours on HalfCheetah-v2).  If combined with CDPO, the running time of the MBRL algorithm is likely to be twice as long. Is it reasonable to take such a long time to obtain an insignificant performance improvement?
4. The benchmark environment for experiments is too simple. Only HalfCheetah is a slightly more complex environment. The authors did not conduct experiments on the complex environments commonly used in MuJoCo such as Ant, Walker2d and Humanoid.
5. The baseline model-based methods (DPI, SLBO) that the author chose to compare in section 6.2 are too old, and these methods are not the current SOTA algorithms. My suggestion is to compare with MBPO [1] and VaGraM [2].

[1] Janner, Michael, et al. "When to trust your model: Model-based policy optimization." Advances in Neural Information Processing Systems 32 (2019).
[2] Voelcker, Claas A., et al. "Value Gradient weighted Model-Based Reinforcement Learning." International Conference on Learning Representations. 2021.

---

> ### Author Response · Authors · 2022-08-02
> **Initial response to Reviewer NcmH (Part 1/2)**
>
> We thank the reviewer for identifying our technical contributions. The valuable comments have helped us improve our submission (marked blue in the revision). The reviewer’s primary concern seems to be the experimental results. Below are our specific responses to the concerns raised by the reviewer:
>
> ---
>
> #### **1. More detailed analysis of the over-exploration issue.**
> Due to space constraints, the theoretical support and experimental evidence were deferred to the Appendix. Following the reviewer's suggestion, we move part of them to the main paper and add more discussions (marked in blue):
>
> - The added **Theorem 3.1** indicates that the eluder dimension of nonlinear models cannot be polynomially bounded, a result provided by previous work. Therefore, additional complexity is hidden in the eluder dimension, e.g. when $\varepsilon = T^{-1}$, regret $O(\sqrt{d_E T})$ contains $d_E = \Omega(T^{d-1})$ and is no longer sublinear in $T$, which is the underlying reason for the over-exploration issue.
> - For experiments, we first add **Section 6.1** to better understand the difference between previous provable algorithms and the proposed one. We highlight our observation here: CDPO gives more accurate and certain estimations _only_ for the optimal actions (still has better regret), while PSRL puts efforts into trying suboptimal actions. This verifies the potential over-exploration issue in PSRL: as long as the uncertainty contains unrealistically large values, PSRL agents can perform uninformative exploration by acting suboptimally according to an inaccurate sampled model. In contrast, CDPO replaces the sampled model with a _stable_ mean estimate and cares about the _expected_ value, thus avoiding such pitfalls.
> - Besides, we add the plot of uncertainty elimination in **Appendix F.3**. The slower uncertainty elimination rate of previous provable algorithms indicates their less informative exploration steps.
>
> ---
>
> #### **2. Choice of trust-region hyperparameter.**
> - The reviewer is correct that the trust-region hyperparameter $\eta$ controls the degree of (conservative) exploration. For this reason, we test a constant $\eta$ and a time-decaying $\eta$. But we are _not_ concluding that a time-decaying $\eta$ performs worse and contradicts our claim. Instead, our results show the robustness of the proposed framework since both of the settings achieve similar asymptotic values with similar timesteps to converge, except that the convergence of a constant $\eta$ is faster in half-cheetah during the first half of training steps. Partial reasons can be the challenge to determine the best range of $\eta$.
> - As per the reviewer's suggestion, we conduct additional experiments on the choice of $\eta$. The results are shown in **Figure 5(b)**. Specifically, we find that changing $\eta = (0.2, 15$%) to $\eta = (0.15, 10$%) gives better performance at the initial stage of training, where $(0.2, 15$%) represents $\eta=0.2$ when initialized and decayed by $15$% every 100 iterations.
> - It is possible to fine-tune the setting of $\eta$ for better performance. However, simple designs (such as constants) are preferable both for training in various environments and for future works. This is also the reason that previous trust-region designs (e.g. DPI [4]), which share similarities with CDPO to control the degree of exploration, adopt the naive constant constraint.
>
> ---
>
> #### **3. Long running time.**
> - Although our dual framework has two update procedures in each iteration, it does **not** indicate the running time is also twice longer since the running time also depends on the number of iterations. Our algorithm has higher sample efficiency and requires a much smaller number of iterations to reach the same performance. Therefore, although the one-iteration running time can be larger, fewer iterations will lead to a smaller total running time.
> - In **Appendix F.2**, we provide a comparison table of sample efficiency and running time for different model-based algorithms. We can observe that CDPO has higher sample efficiency and shorter running time, e.g., MBPO needs $2.7$ times more samples to reach the same performance as CDPO in half-cheetah and requires $50$% more running time.
>
> ---
>
> #### **4. Performance in larger environments.**
> - As suggested by the reviewer, we add additional experiments in the Ant task in our revision (**Fig. 3(d)** and **4(d)**).

---

> > ### Author Response · Authors · 2022-08-02
> > **Initial response to Reviewer NcmH (Part 2/2)**
> >
> >
> > #### **5. Comparison with SOTA.**
> > - We would like to first emphasize that our primary goal is to address the principled over-exploration issue in _provably efficient_ algorithms (especially PSRL and OFU-RL), which are typically evaluated in _tabular settings_ in previous work (e.g. HUCRL [1] is one of the first provable algorithms to test in MuJoCo). Our contribution is a dual framework that provides guarantees in nonlinear settings where two basic designs are suggested: choosing a _stable_ reference model and maximizing the _expected_ value. We are not claiming to beat all SOTA algorithms with sophisticated designs (although we have included comparisons with more than $8$ baselines in our initial submission). Instead, our implementation is _pretty naive_: we use least-square models with Dyna as the default policy solver. We expect our simple yet effective framework to inspire more future studies on provable algorithms in nonlinear settings and complex environments.
> > - As suggested by the reviewer, we add the performance comparison of MBPO [2] and VaGraM [3] in **Figure 4**. Due to the time limit, in the current revision, we report the results of VaGraM in the inverted pendulum, ant tasks, and MBPO in the inverted pendulum, half-cheetah, ant tasks.
> > - It is also worth noting that specific designs are _complementary to our work_ and can be integrated with our framework (c.f. Appendix E for several algorithmic instantiations). For instance, MBPO[2] and our default Dyna solver both apply model-free algorithms with model-generated samples (i.e. CDPO without the conservative update step), and it might also be an option to leverage value-equivalent models and combine with VaGraM [3].
> >
> > ---
> >
> > We hope the reviewer can consider raising the score if we resolved the reviewer's concerns. We would be happy to have further discussions if the reviewer has any additional questions or comments.
> >
> >
> > ---
> >
> > [1] Curi et al. "Hallucinated-UCRL." NeurIPS 2020.\
> > [2] Janner et al. "Model-Based Policy Optimization." NeurIPS 2019.\
> > [3] Voelcker et al. "VaGraM." ICLR 2022.\
> > [4] Sun et al. "DPI." NeurIPS 2018.

---

> > > ### Comment · Reviewer_NcmH · 2022-08-08
> > > **Response**
> > >
> > > Thank you for your hard working! My concerns have been addressed. While I think the baseline comparison should include more environments, I will increase my score from 3 to 5.
> > >
> > > Best

---

> > > > ### Author Response · Authors · 2022-08-08
> > > > **Response to reviewer NcmH**
> > > >
> > > > Thank you for helping us improve the paper and for updating the score! We really appreciate your suggestions and will continue polishing our paper.

---

> ### Author Response · Authors · 2022-08-06
> **Follow up before the discussion period closes**
>
> We want to thank the reviewer NcmH again for the valuable feedback, which has helped us improve our submission in the updated revision. As we are halfway through the discussion period, we would like to ask whether our revision and responses clarify the questions raised in your initial reviews. We are happy to provide any further clarification and discussion.

---

### Author Response · Authors · 2022-08-02
**Author response to all reviewers**

We thank the reviewers for their valuable comments. We are excited that the reviewers identified the novelty of our technical contributions (Reviewer NcmH, 2poR, M8qo, q3JM), found our analysis sufficient or comprehensive (Reviewer NcmH, 2poR, M8qo, q3JM), appreciated the importance of the studied problem (Reviewer 2poR, M8qo), and acknowledged our experimental validations (Reviewer 2poR, M8qo).

---

In our updated revision, we provide major improvements by clarifying _all_ raised questions. Notable changes in the main paper and the Appendix are enumerated below (marked blue in the revision).

- To better understand the limitation of previous exploration mechanisms and to motivate how the proposed method works, we move part of our existing results from Appendix to **Section 6.1** and add more discussion.
- We move **Theorem 3.1** (a result established in previous work) from Appendix to Section 3 as more straightforward support of our analysis of the over-exploration issue.
- We add the higher-dimensional Ant experiment in **Section 6.3**.
- We provide the uncertainty elimination curve in **Appendix F.3** and the comparison table containing the sample efficiency & running time in **Appendix F.2**.

We also add the requested results and detailed discussions posed by individual reviewers. Please see our response below.

---

### Author Response · Authors · 2022-08-07
**Message to Reviewer NcmH and q3JM**

Dear Reviewer **NcmH** and **q3JM**,

We greatly appreciate your insightful comments on our paper. As the discussion period is coming to an end, we respectfully look forward to following up to see if our response has addressed your concerns or if you have any further questions. Here, we provide a brief summary of how we addressed the main concerns of Reviewer **NcmH** and **q3JM** about our motivation and experimental validations:

---

**Reviewer NcmH:** We thank the reviewer for the helpful suggestion.
- We move part of the results from the Appendix to the main body: we add Theorem 3.1 to **theoretically support** our analysis of the over-exploration issue and add Section 6.1, Appendix F.3 as **empirical evidence** of the over-exploration issue and how the proposed method overcomes it.
- We add all the requested experiments:  **(1).** We add evaluations in the Ant task; **(2).** We compare with two more SOTA baselines MBPO and VaGraM; **(3).**  We conduct additional experiments on the choice of hyperparameter with additional discussions; **(4).** The concerned running time will not be twice longer since CDPO needs fewer iterations to converge. We provide a table in Appendix F.2 for a clear comparison.

We would be very glad if the reviewer could reconsider the score of our work. Thank you!

---

**Reviewer q3JM:** We thank the reviewer for the positive feedback.
- For a clearer motivation, we move part of the results from the Appendix to the main body: we add **Theorem 3.1** to support our analysis of the over-exploration issue and add **Section 6.1** to better understand the limitation of previous algorithms and how the proposed method benefits exploration.
- We also add the additional evaluations requested by the reviewer:  **(1).** We add evaluations in the Ant task; **(2).** We add the comparison table of the sample efficiency improvement ratio.

We would be delighted if the reviewer is willing to increase the score if our response has addressed your question. Thank you!

---

Please let us know if you have any further questions and suggestions.

Best Regards,

Author of Paper4255

---

### Meta-Review · Area_Chair_wjsz · 2022-08-26

**Recommendation:** Accept
**Confidence:** Certain

**Metareview:**

Reviewers all appreciated the authors effort on adding additional experiments and revising the draft during the rebuttal phase, and the reviewers are in general satisfied by the authors response. The reviewers agree that the paper has a solid contribution to MBRL.

**Award:**

No

---

### Decision · Program_Chairs · 2022-09-14

Accept